# BATF3 regulates differentiation of CD8+ T lymphocytes and memory differentiation program

Koji Umemura[1],*, Yasuhiro Kojima[2],*, Jakrawadee Julamanee[1,3], Yusuke Okuno[4,5], Yuki Takeuchi[1], Fumiya Ohara[1], Shihomi Kuwano[1], Yoshitaka Adachi[1], Ryo Hanajiri[1], Seitaro Terakura[1], Hitoshi Kiyoi[1]

**CD8+ T lymphocytes differentiate from effector to memory cells after antigen clearance, with prolonged IL-2 production characterizing functional cytotoxic T lymphocytes (CTLs). To identify transcription factors associated with sustained IL-2 production, we compared influenza virus–specific and cytomegalovirus-specific CTLs, identifying Basic Leucine Zipper ATF-Like Transcription Factor 3 (BATF3) as a key candidate. BATF3 overexpression significantly enhanced cell proliferation in both virus-specific CTLs and CD19 chimeric antigen receptor T (CAR-T) cells while reducing cytokine production. Among AP-1 transcription factor family members, BATF and BATF3 demonstrated similar functions during effector phase expansion, but BATF3 exhibited distinct roles in promoting memory cell formation. ATAC-seq analysis revealed that BATF3 overexpression dynamically regulates chromatin accessibility, affecting diverse cellular processes including cytoskeletal organization, metabolic pathways, and survival signaling. BATF and BATF3 showed comparable kinetics until peak expansion, but BATF3 specifically facilitated the transition from effector to memory phase, up-regulating memory-associated genes while down-regulating exhaustion markers. These findings establish BATF3 as a master regulator of CD8+ T-cell fate determination through chromatin remodeling, offering therapeutic targets for enhancing CAR-T cell persistence in immunotherapy.**

## Introduction

CD19 chimeric antigen receptor T (CAR-T) cells have demonstrated remarkable clinical outcomes against B-cell malignancies, such as B-cell acute lymphoblastic leukemia (B-ALL) (1, 2, 3, 4), diffuse large B-cell lymphoma (DLBCL) (5, 6, 7, 8), chronic lymphocytic leukemia,

and others (9, 10, 11, 12). Surprisingly, high initial remission rates are shown. However, half of the B-ALL patients who exhibited initial remission still resulted in clinical relapse during the first 6 mo after CAR-T infusion (2, 3, 4). To improve clinical outcomes including other CAR-T therapies, CAR-T in vivo persistence is a hallmark of excellent quality of CAR-T cells (13, 14, 15). Accordingly, improving CAR-T expansion and persistence is a current challenge to achieve better long-term survival.

To enhance the expansion and persistence of CAR-T cells, virus-specific memory T cells can be a model to emulate because virus-specific memory T cells can mount immune responses more rapidly and robustly than after initial encounter with the pathogens (16, 17). A possible strategy involves the screening of candidate genes associated with memory T-cell formation and further genetic modification into CAR-T cells. Candidate genes involved in the formation of memory T cells have become clearer through experiments using small animal models (18, 19). Because it is possible that the important genes associated with memory T-cell formation differ between small animals and humans, we sought to adopt human infection models with established virus-specific memory T cells in humans. To this end, influenza virus (Flu) is a representative of acute infection models, and cytomegalovirus (CMV) and Epstein–Barr virus (EBV) are representative of latent infection ones (16, 20, 21, 22). In CMV and EBV infections, repeated and more frequent antigen exposures by the virus reactivation made more highly differentiated phenotype T cells (23, 24). These failed to produce IL-2 and occasionally even TNF-α and IFN-γ among virus-specific CD8+ T cells (25, 26, 27). Accordingly, we started with candidate gene screening by comparing Flu-specific T cells and CMV-specific T cells in terms of the IL-2–producing capacity as a surrogate endpoint for earlier effector cells to differentiate into memory T cells, and focused on *Basic Leucine Zipper ATF-Like Transcription Factor 3* (*BATF3*) as a candidate gene.

In this study, we focused on *BATF3* after screening an infection model using human cells. The overexpression (OE) of *BATF3* in human virus-specific T cells and CD19 CAR-T cells improved their

---

[1]Department of Hematology and Oncology, Nagoya University Graduate School of Medicine, Nagoya, Japan [2]Laboratory of Computational Life Science, Research Institute, National Cancer Center, Tokyo, Japan [3]Hematology Unit, Division of Internal Medicine, Faculty of Medicine, Prince of Songkla University, Songkhla, Thailand [4]Department of Clinical Development, Nagoya University Graduate School of Medicine, Nagoya, Japan [5]Department of Virology, Nagoya City University Graduate School of Medical Sciences and Medical School, Nagoya, Japan

Correspondence: tseit@med.nagoya-u.ac.jp; yakojim@ncc.go.jp
*Koji Umemura and Yasuhiro Kojima contributed equally to this work

 

proliferative capacity. Meanwhile, CD 19 CAR-T cells overexpressing *BATF3* decreased the production of cytokines such as IFN-γ and IL-2. ATAC-seq showed that BATF and BATF3 work similarly in the effector phase, but BATF3 is mainly responsible for memory cell formation in the later phase.

# Results

## Repetitive stimulation of virus-specific CD8⁺ T lymphocytes (CTLs) and those responses

The immune response to cleared acute infections without persistent antigen (such as influenza) is characterized by the long-term presence of true quiescent memory cells, whereas the chronic phase of persistent infections (CMV, EBV) is characterized by the presence of effector phenotypes with varying degrees of more highly differentiated cells (28, 29). To investigate the differences between less differentiated memory T cells and highly differentiated memory T cells, we used Flu-specific T cells as less differentiated memory T cells and CMV-specific T cells as a model of highly differentiated T cells. We applied repetitive Ag peptide stimulation until day 30 (Fig 1A) and analyzed the responses in terms of proliferation and cytokine production (Fig S1A). In terms of tetramer+ T-cell proliferation, CMV-specific CTLs peaked earlier than Flu-specific CTLs. The cumulative fold expansion from culture initiation was greater in CMV-CTLs than in Flu-CTLs (Fig 1B and C). When we examined the surface phenotype of tetramer-positive cells during the culture period, Flu-specific CTLs were predominantly CD45RA+/CD62L+ naive cells before stimulation, and CD45RA-CD62L+ central memory T cells on day 10 (Figs 1D and E and S1B). Although the cytokine production capacity of CMV-CTL decreased during the culture period, both in IFN-γ and in IL-2, that of Flu-CTL was maintained during the culture period, particularly in IL-2 production (Figs 1F and G and S1C and D). Taken together, these data suggest that Flu-CTL seemed to be less differentiated T cell in humans, which maintained its cytokine production capacity.

## RNA-sequencing screening revealed transcription factors (TFs) associated with earlier memory phenotype

To unveil the transcriptional differences between CMV-CTL and Flu-CTL, we harvested CMV-CTL and Flu-CTL on days 20 and 30 from three different donors, respectively, extracted RNA, and performed RNA-seq. Paired CMV-CTL and Flu-CTL cultures were generated from each of the three donors. Given the inherent donor-to-donor variability in human T-cell responses, our paired analysis approach with a permissive threshold (FDR < 0.1) was designed to prioritize sensitivity in identifying candidate genes for subsequent functional validation, accepting a higher risk of false positives during this discovery phase. On comparing day 20 and day 30 CMV-CTL with Flu-CTL, we identified 258 differentially expressed genes (DEGs) from day 20 and 204 DEGs from day 30, which showed an adjusted *P*-value less than 0.1 and log₂ (fold change) > 1 or < −1. Of these genes, we searched the list to obtain the genes that exhibited

expression on T lymphocytes in the database and obtained candidate gene lists consisting of 63 genes from day 20 and 47 genes from day 30 comparisons (Fig 2A). A volcano plot demonstrating the differentially expressed genes between CMV-CTL and Flu-CTL on days 20 and 30 is shown (Figs 2B and S2). The expression of *JUN* and *EOMES* was higher in CMV-CTL. Meanwhile, we observed several genes with greater expression in Flu-CTL. This included *HAVCR2*, which encodes T-cell immunoglobulin and mucin domain containing-3 (TIM-3), and *BHLHE40*, which has been reported to regulate T-cell effector function and differentiation (30, 31). When we plotted the adjusted *P*-value of significant genes from days 20 to 30, a considerable number of genes lost significance on day 30 (Fig 2C). Accordingly, we focused on day 20 comparisons and selected five candidate genes that demonstrated higher expression levels in Flu-CTL: *HTATSF1*, *IKZF4*, *BATF3*, *PHTF2*, and *ID2*. Among these five genes, *BATF3*, *IKZF4*, and *ID2* are established TF/regulators with relatively well-characterized roles in T-cell biology, whereas *PHTF2* and *HTATSF1* represent less well-characterized factors.

## BATF3 overexpression maintains CAR-T cell survival while lowering cytokine production

To further assess the functions of candidate TF genes under Ag-specific stimulation, we transduced candidate TF and CD19 CAR into CD8⁺ T cells. CD19 CAR consisted of FMC63-derived scFv, CD28 transmembrane and costimulatory domain, and CD3zeta, along with the tEGFR gene marker. The TF gene construct consisted of a TF ligated to an EGFP gene marker via the IRES sequence (Fig 3A). CD8⁺ T cells were stimulated with CD3/CD28 beads, retrovirally transduced with CD19 CAR and TF on day 3, purified by EGFR selection, and further stimulated with K562-CD19 every 10 d (Fig 3B). Although we observed some variability in transduction efficiency among the TFs, we successfully established CD19 CAR-T cells expressing each TF (Fig 3C). During repeated stimulation with K562-CD19, CD19 CAR-T cells with BATF3 maintained a significantly higher proportion of TF (EGFP+). Meanwhile, CD19 CAR-T cells with other TFs showed a similar percentage of TF+CD19 CAR-T cells, whereas only BATF3+CD19 CAR-T cells demonstrated the progressive enrichment of transduced cells (Fig 3D). BATF3+CD19 CAR-T cells showed significantly lower IFN-γ– and IL-2–positive cell frequency after K562-CD19 simulation #3 (Fig 3E and F). These results suggest that BATF3-OE gave the CD19 CAR-T cells an anti-apoptotic tendency while maintaining lower but constant cytokine production.

*BATF3*, a member of the AP-1 TF family, is required for the differentiation of type 1 conventional dendritic cells in mice (32). *BATF3* is involved in the proliferation and maintenance of memory cells in mouse CD8⁺ T cells by mediating apoptosis resistance (33). Furthermore, there is another AP-1 TF, *BATF*, which regulates CD8⁺ T-cell proliferation and differentiation (34, 35), as well as exhaustion (36). Both BATF and BATF3 have shown their association with the process of T-cell differentiation and memory generation (33, 37, 38). Therefore, BATF family TF attracts considerable attention to illuminate their functions in T cells and also those potential to modulate CAR-T cell functions.

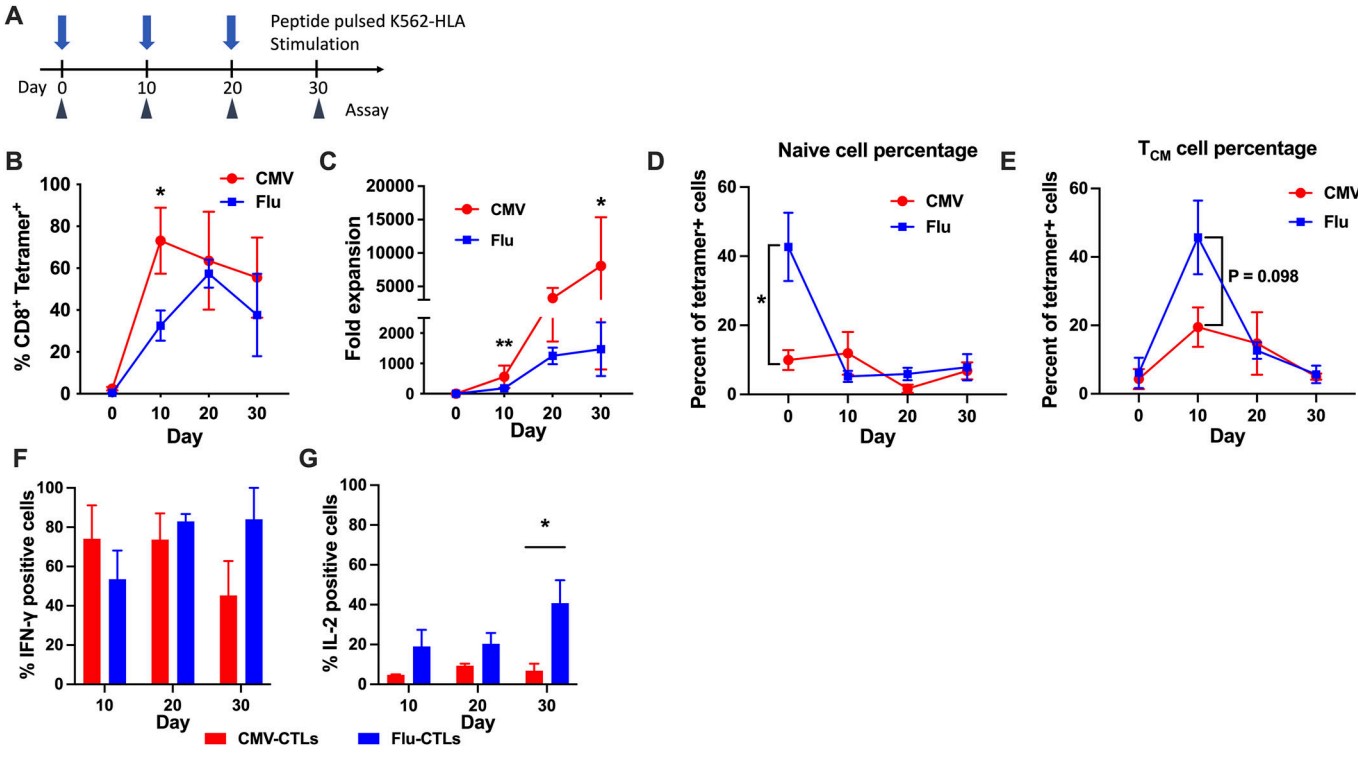

**Figure 1. Repetitive stimulation of virus-specific CTLs demonstrates differential growth and cytokine production patterns.**
**(A)** Schematic of repetitive stimulation experiments. PBMCs isolated from healthy donors were cultured with viral peptides (5 µg/ml) supplemented with rhIL-2 50 IU/ml every 3 d. On day 10 and thereafter, the CTL lines were further stimulated with peptide-pulsed HLA-transduced K562 cells every 10 d. **(B)** Percentages of CD8+ tetramer+ cells during repetitive stimulation cultures. Virus-specific CTLs were stimulated with peptide-pulsed HLA-K562 at a 1:1 ratio. The data are presented as the mean ± SEM (*t* test; *P < 0.05). **(C)** Fold expansion of CD8+tetramer+ cells during repetitive stimulation cultures. **(B, C)** Live cells and lymphocytes were gated, and then, CD8+/tetramer+ cells were gated to quantify CD8+tetramer+ cells (B, C). **(D, E)** Percentages of naïve and central memory T cells. **(D, E)** Live cells and lymphocytes were gated, and then, CD8+/tetramer+ cells were gated to apply CD45RA/CD62L flow plots (D, E). **(F, G)** Percentage of intracellular IFN-γ– and IL-2–positive cells at each time points, respectively. Virus-specific CTLs were stimulated at 1:1 ratio with peptide-pulsed HLA-K562 at each indicated time points, and intracellular cytokine was analyzed after 4 h of culture. All data were pooled from three different donors and are presented as the mean ± SEM. All three donors provided one CMV-CTL line and one Flu-CTL line, respectively. **(B, C, D, E)** Significance levels are shown only for the comparisons with significance (*t* test for (B, C, D, E); *P < 0.05, **P < 0.01).

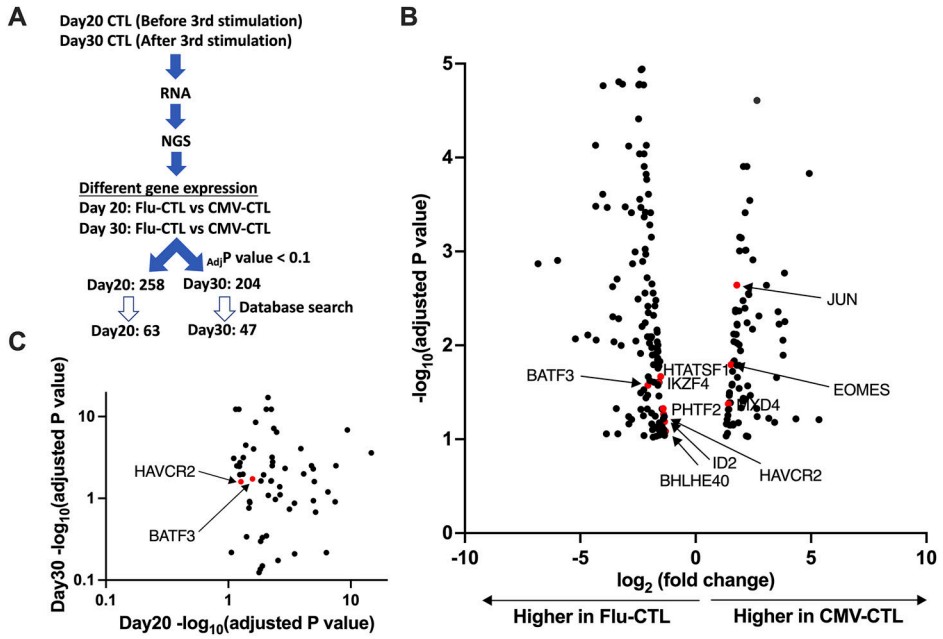

**Figure 2. Differential gene expression determined by RNA sequencing between CMV-CTL and Flu-CTL.**
**(A)** Outline of experiments. NGS, next-generation sequencing; $_{Adj}$P-value, adjusted P-value. **(B)** Volcano plot showing the differentially expressed genes between CMV- and Flu-specific CTLs on day 20. Genes with adjusted P-value less than 0.1 are shown. **(C)** Adjusted P-value from day 20 and day 30 comparisons. Of 258 significant genes showing adjusted P-value less than 0.1 from day 20 comparison, adjusted P-value from day 30 comparison was available only in 56 genes. Adjusted P-values available of both data were plotted.

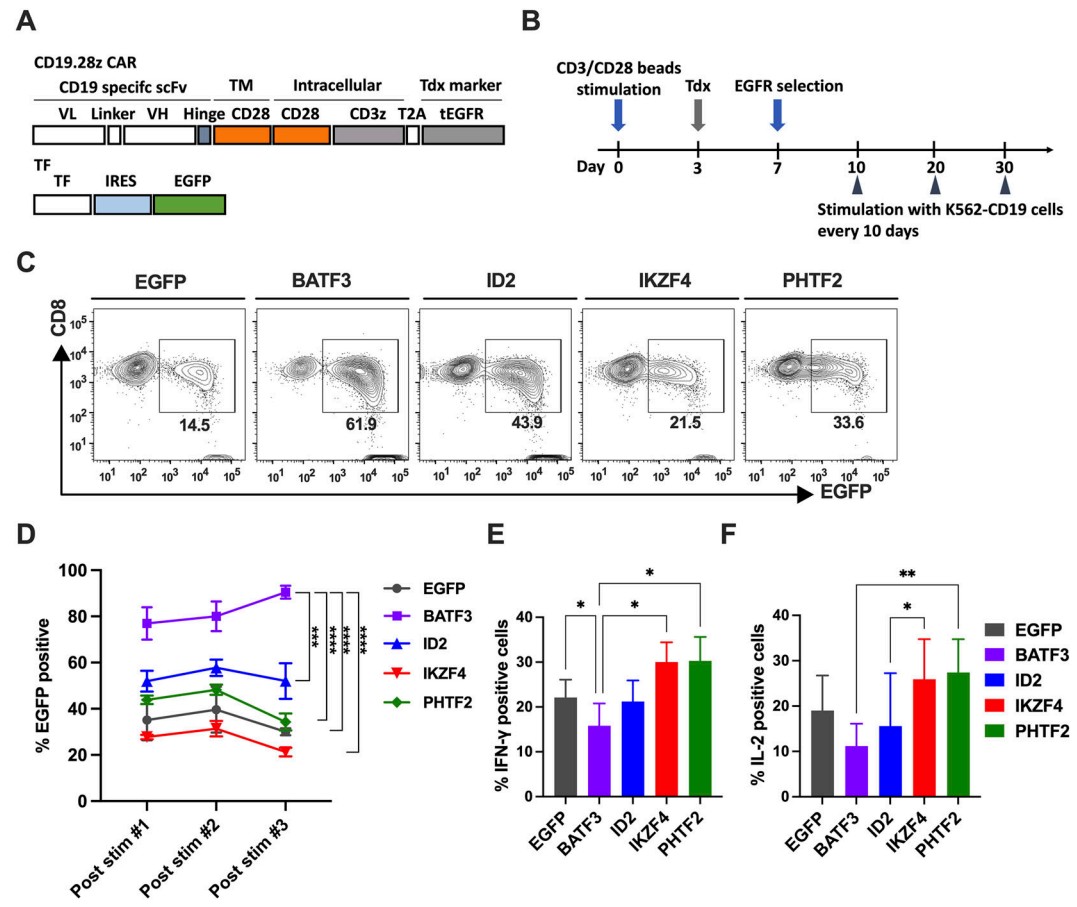

**Figure 3.  BATF3 transduction to CD19 CAR-T cells maintains high CD8+ T-cell viability with lower cytokine production capacity.**
**(A)** Schematic of CD19 CAR and TF constructs. A CD28 costimulatory domain was fused into anti-CD19scFv-H-CD28 followed by CD3ζ and tEGFR. VL, light chain variable fragment; VH, heavy chain variable fragment; H, IgG4 Fc-derived hinge domain (12 amino acids); TM, transmembrane domain; tEGFR, truncated EGFR. The gene construct for TF was fused with EGFP via IRES sequence. **(B)** Outline of experiments. CD8+ T cells were stimulated with CD3/CD28 beads on day 0. Then, candidate TFs and CD19 CAR were simultaneously transduced on day 3; CD19 CAR+ cells were purified by EGFR selection on day 7 and further stimulated with K562-CD19 every 10 d. Tdx, transduction. **(C)** Representative flow plots of TF gene transduction after purification of CD19 CAR-T cells by EGFR selection. More than 90% of cells were CD19 CAR-T cells after EGFR selection. The rectangle region is gated for EGFP+ (TF+) cell frequency. **(D)** Percentage of EGFP-positive cells during repetitive stimulation cultures. TF-transduced CD19 CAR-T cells were stimulated with γ-irradiated K562-CD19 at a 1:1 responder-to-stimulator ratio every 10 d. **(E, F)** Intracellular cytokine staining for IFN-γ (E) and IL-2 (F) post-third stimulation. TF+CD19 CAR-T cells were stimulated with K562-CD19 cells at an E:T ratio of 1:1 for 4 h, and then permeabilized and stained for intracellular IFN-γ and IL-2. **(D, E, F)** Data were pooled from four different donors and are presented as the mean ± SEM in (D, E, F). Significance levels are shown only for the comparisons with significance (one-way ANOVA, *P < 0.05, **P < 0.01, ***P < 0.001, ****P < 0.0001).

## AP-1 family TF in virus-specific CTLs

The AP-1 TF family, such as *JUN* and *BATF*, plays important roles in the development of effector and memory T-cell formation, respectively (37, 39). Because *BATF3* also belongs to the AP-1 TF family, we attempted to reanalyze whether AP-1 TF family members play a significant role in virus-specific T-cell development or maintenance. We selected 22 AP-1 family TFs (40) and reanalyzed the comparison data between day 20 CMV-CTL and day 20 Flu-CTL and found that *FOSL1* and *BATF* were also weakly significant (Fig 4A). We further verified the time course expression of these TFs (*JUN*, *BATF3*, *BATF*, and *FOSL1*) using qRT-PCR. The virus-specific CTL frequencies of CMV-CTL and Flu-CTL were in the same range, as previously shown (Fig 1B). In both Flu-CTL and CMV-CTL, we confirmed that BATF3 and BATF expression increased during the repetitive stimulation culture period. Meanwhile, JUN maintained

similar expression levels, and FOSL1 showed an earlier decrease by day 20 and a later surge after day 30 (Fig 4B–D).

We further investigated the potential effects of TF overexpression in virus-specific CTLs. Starting with PBMCs stimulated with the Ag peptide, we applied lentiviral transduction of each TF on day 3, and repetitive stimulations were undertaken thereafter. After each course of stimulation, CD8+ cells were selected using immunomagnetic beads, and RNA was extracted and subjected to qRT-PCR (Fig 4E). Representative flow plots of CMV-CTLs and Flu-CTL demonstrated that BATF and BATF3 contributed to the accumulation of CTL (Figs 4E and S3, respectively). The number of EGFP+ (TF+) cells increased with all three TF transductions in Flu-CTL (Fig 4F). Meanwhile, only BATF and BATF3 transductions resulted in an increase in EGFP+ cell numbers in CMV-CTL (Fig 4G). Because the TF+/tetramer+ cells and TF−/tetramer+ cells were cultured in the same well, we calculated the TF+/TF− ratio. For both Flu-CTL and

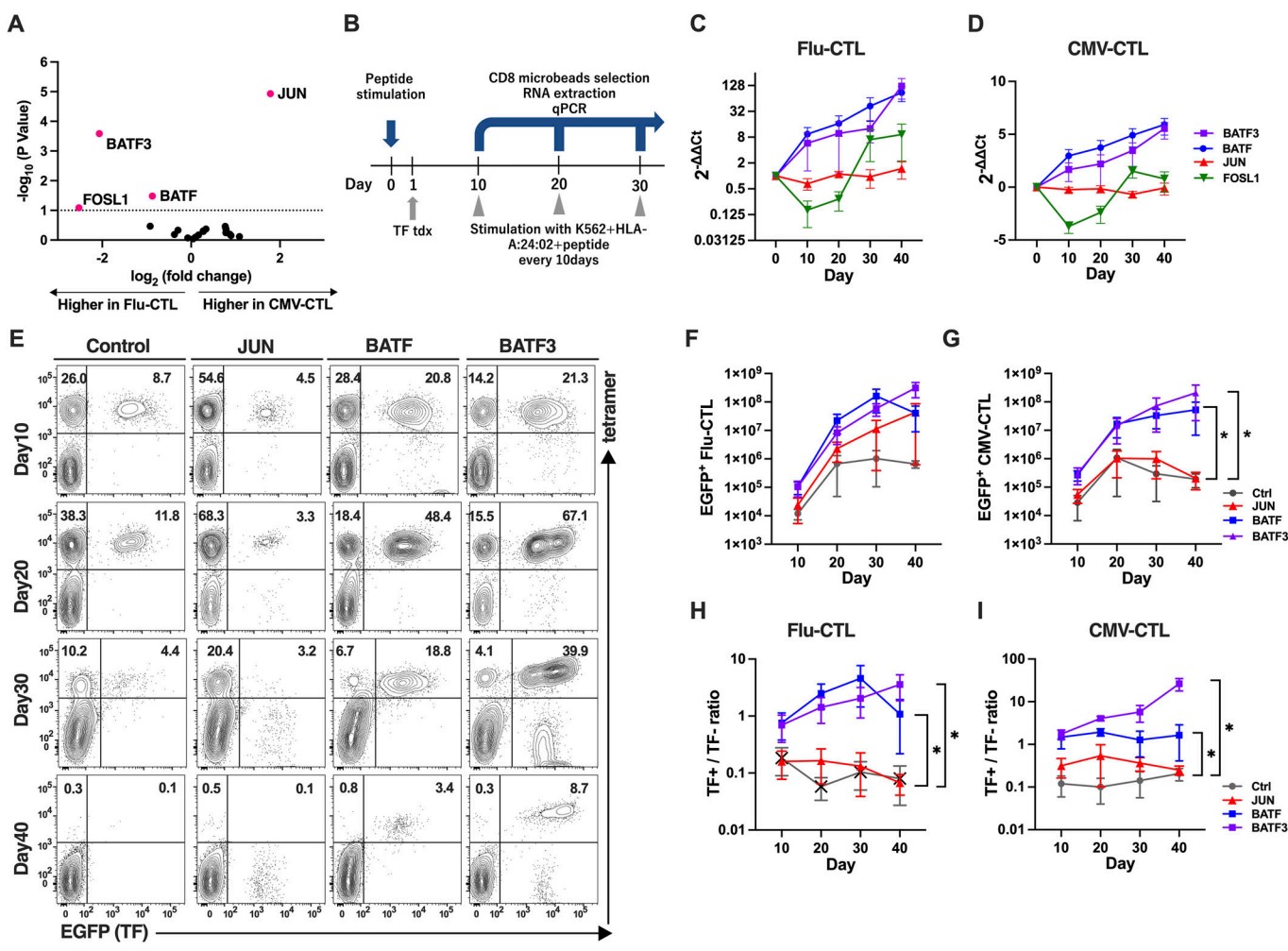

**Figure 4. AP-1 family TF expression of and transduction to virus-specific CTLs.**
**(A)** Volcano plot showing differentially expressed genes of all AP-1 TF family between CMV- and Flu-specific CTLs on day 20. The data were reanalyzed using the same experiments shown in Fig 2A. Genes with *P*-values less than 0.1 are colored in red. **(B)** Schematic of experiments. PBMCs were stimulated with 5 μg/ml antigen peptide on day 0. Candidate TFs were lentivirally transduced on day 1 and stimulated with peptide-pulsed K562-HLA every 10 d. CD8+ cells were purified on days 10, 20, and 30, and RNA was extracted for qRT-PCR analysis. Tdx, transduction. **(C, D)** Time course of AP-1 TF family expression in Flu-CTL (C) and CMV-CTL (D). Data were obtained by qRT-PCR and normalized to the expression of *HPRT*, a housekeeping gene. **(E)** Representative flow plots of TF+CMV-CTLs. Lymphocytes and CD8+-gated cells are shown. The data are representative of five independent experiments involving five donors. Numbers represent the percentage of each fraction of gated cells. **(F, G)** Time course of TF+(EGFP+) tetramer+ cell count for Flu-CTL (F), and CMV-CTL (G). **(H, I)** Time course of ratio of TF+ tetramer+ cell fraction to TF- tetramer+ cell fraction for Flu-CTL (H) and CMV-CTL (I). **(F, G, H, I)** Significance levels are shown only for comparisons with significance (one-way ANOVA; *$P < 0.05$) in (F, G, H, I).

CMV-CTL, BATF and BATF3 showed significantly higher TF+/TF− ratios than control and JUN-transduced cells (Fig 4H and I). Taken together, BATF and BATF3, but not JUN transduction, added certain anti-apoptotic traits to virus-specific T cells or increased proliferation capacity. This effect was particularly evident in CMV-CTLs.

### AP-1 TF family transduction in CAR-T cells

To examine the effect of TF-OE on CAR-T cells, we lentivirally transduced TF on day 1 and retrovirally transduced CD19 CAR on day 3 into CD8+ cells (Fig 5A). After EGFR selection, CD19 CAR-T cells with BATF or BATF3 showed a higher percentage of EGFP+ cells than CD19 CAR-T cells with control or JUN (Fig 5B and C). We observed higher EGFP+ cell percentages in BATF- and BATF3-transduced

CD19 CAR-T cells throughout the culture period. BATF3+ CAR-T cells showed a higher EGFP+ population than BATF+CAR-T cells (Fig 5C). We observed a significantly higher fold expansion during repetitive culture of BATF3+CD19 CAR-T cells (Fig 5D). In contrast to proliferation, CD19 CAR-T cells expressing BATF3 showed a significantly lower cytokine production after repetitive stimulation (Fig 5E and F). CD19 CAR-T cells with BATF and BATF3 demonstrated significantly lower $T_{CM}$ and higher $T_{EM}$ proportions after repetitive stimulation (Fig 5G–I). Each CD19 CAR-T cell was stimulated with CD19-positive tumor cell lines, and the percentage of live cells was measured after 24 h. Only BATF3-OE CD19 CAR-T cells showed a significantly higher percentage of live cells than control CD19 CAR-T cells (Fig 5J). Surface phenotype analysis after three repeated stimulations demonstrated that the expression levels of TIM-3 and

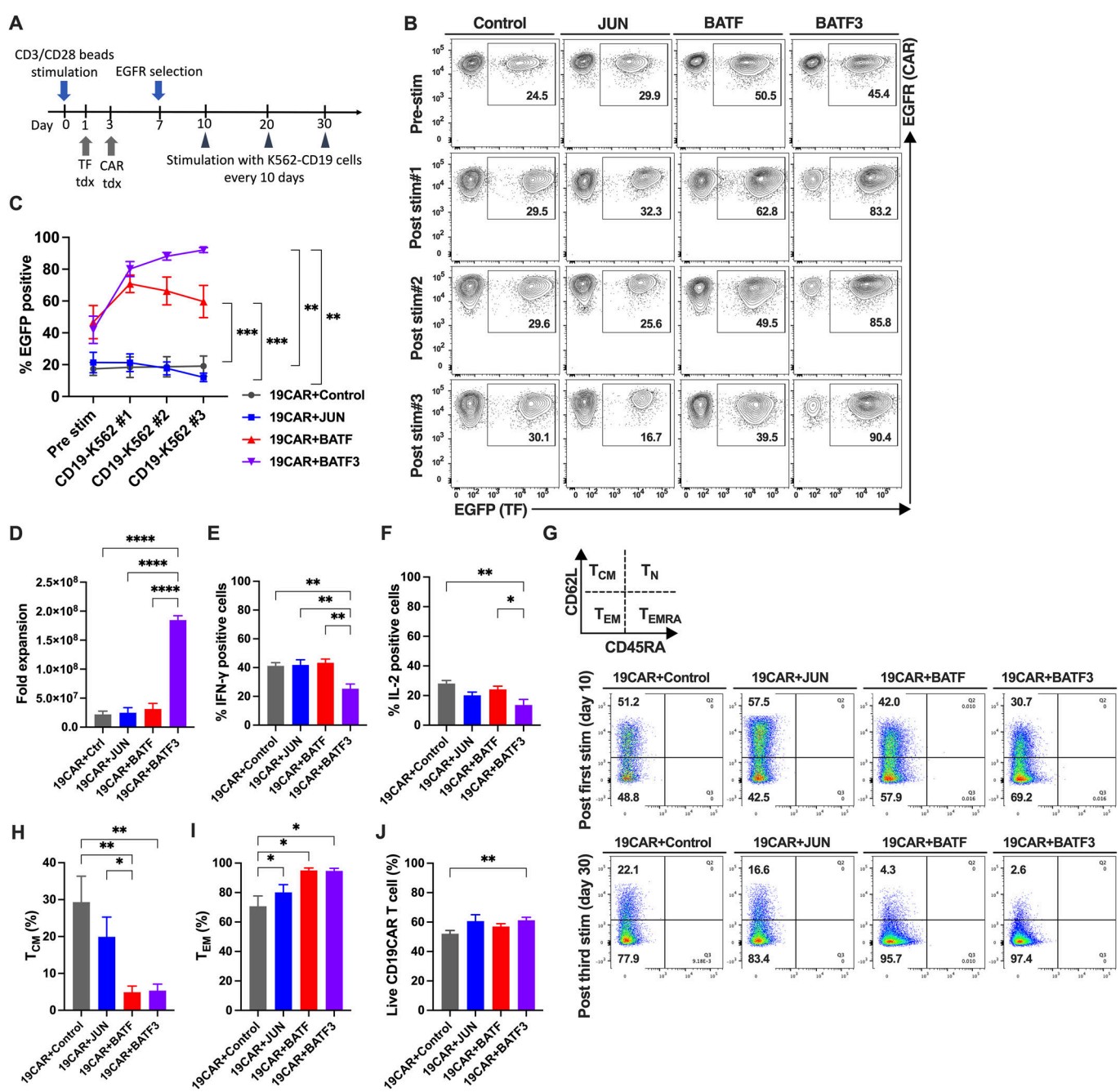

**Figure 5. BATF3 but not other AP-1 family TF transduction to CD19 CAR-T cell mediates prosurvival effect with maintenance of lower cytokine production capacity.**
**(A)** Schematic of experiments. CD8+ T cells were stimulated with CD3/CD28 beads on day 0. Then, candidate TFs and CD19 CAR were sequentially transduced lentivirally on day 1 and retrovirally on day 3, respectively. CD19 CAR+ cells were purified by EGFR selection on day 7 and further stimulated with K562-CD19 every 10 d.
**(B)** Representative flow plots of TF gene transduction after purification of CD19 CAR-T cells by EGFR selection. Displayed numbers are frequencies of the EGFP+ region.
**(C)** Time course of TF+(EGFP+) CD19 CAR-T percentages during repetitive stimulation culture. **(D)** Fold expansion during the repetitive CD19 stimulation assay. **(E, F)** Intracellular cytokine staining for IFN-γ (E) and IL-2 (F) post-K562-CD19 stimulation #3. TF+CD19 CAR-T cells were stimulated with K562-CD19 cells at an E:T ratio of 1:1 for 4 h, and then permeabilized and stained for intracellular IFN-γ and IL-2. **(G)** Representative flow plots of TF-transduced CD19 CAR-T cells after three repetitive stimulations. Live cells and lymphocytes were gated, and then, CD8+/GFP+ cells were gated to show as CD45RA/CD62L flow plots. **(H, I)** T-cell subsets were determined by CD45RA and CD62L expression. **(H, I)** Percentages of CD45RA⁻ CD62L⁺ central memory T (T_CM) (H) and CD45RA⁻ CD62L⁻ effector memory T (T_EM) cells (I) after K562-CD19 stimulation #3. Percentages of TF-positive cells are shown. **(C, D, E, F, H, I, J)** Data were pooled from three different donors and are presented as the mean ± SEM in (C, D, E, F, H, I, J). Significance levels are shown only for the comparisons with significance (one-way ANOVA, *P < 0.05, **P < 0.01, ***P < 0.001, ****P < 0.0001).

LAG3 were similar between BATF3-OE and control groups. However, the expression levels of CTLA-4 and PD-1 were significantly lower in the BATF3-OE group than those in the control (Fig S4A–D).

### BATF3 influences the chromatin accessibility to remodel the CAR-T cell to memory signature

To determine the potential mechanism by which BATF3-OE causes the CD19 CAR-T cells to accumulate in culture, we performed ATAC-seq after CD19 stimulation. After EGFR selection, we stimulated the TF-transduced CD19 CAR-T cells with K562-CD19, harvested the CD19 CAR-T cells before stimulation on days 7 and 14, and performed ATAC-seq. We performed our analysis using the nf-core/atac-seq pipeline (41). PCA did not show a trend of accumulation according to the transduced TF, but rather, the number of days elapsed after stimulation (Fig 6A). PC1 expression was maximized on day 7 and returned to baseline on day 14, indicating an association with effector activity.

Next, we extracted CD8$^+$ population data from the public single-cell transcriptome data (42) and calculated the effector score from the similarity with the naïve–effector transition and the memory score from the similarity with the effector–memory transition (Fig 6B). We converted the ATAC-seq chromatin status data to estimated read counts per gene based on the peak overlap with the gene region and calculated the effector and memory scores for each sample. Effector scores were higher on day 7 and lower on day 14. The memory score increased from days 0 to 7 and from days 7 to 14 in BATF3-OE CAR-T cells (Fig 6C). When each score was plotted over time, the effector score did not differ between the groups. However, the memory score increased only in the BATF3 group on day 14 (Fig 6C and D). To identify the genes that contributed to the memory score, the difference in expression between the control and BATF3 groups on day 14 was plotted along with the contribution of the memory and effector signatures (Fig 6E). We identified 134 genes with significantly high memory score genes and 142 with significantly low memory scores. Among these memory-related gene groups, those with particularly high and low memory correlation scores were plotted (Fig 6F and G). Low memory score genes included BATF and exhaustion-related genes, such as LAG3 and TIGIT. When pathway analysis using Metascape was applied to these gene groups, similar pathways, such as T-cell activation and cytotoxicity, were predominant regardless of whether a low or high memory score was applied (Fig 6H and I). We further observed the memory signature genes up-regulated in the BATF3 group and the effector signature genes down-regulated in the BATF3 group (Fig 6E). The time course of the memory score showed no difference between the BATF-transduced group and the BATF3-transduced group for TIGIT and LAG3, and a decrease at the final time point. CX3CR1, CTSC, and FLNA levels decreased more in the BATF3 group than in the BATF group. Conversely, a few genes were elevated in the BATF3 group and were associated with higher memory scores, including MYOM2 and RAB11FIP1. These were more elevated in the BATF3 group than in the BATF group at the final time point (Fig 6J). Together, these data demonstrate that BATF3-OE dynamically regulates chromatin accessibility to establish a memory-associated transcriptional program, characterized by up-regulation of prosurvival genes and down-regulation of

exhaustion-associated genes, specifically during the transition from effector to memory phase (day 7 to day 14).

### Comparison between BATF3-OE and BATF-OE identifies several candidate TFs

We further compared the BATF3 and BATF groups to identify genes that contribute to the memory score, and the difference in expression between the BATF3 and BATF groups on day 14 was plotted along with the contribution of the memory and effector signatures (Fig 7A). When we focused on TF alone, we identified several candidate TFs within the BATF-higher area (MAF, LEF1, TCF7, HOPX, and MXD4; Fig 7B) and within the BATF3-higher area (LYAR, PRDM1, AKNA, KLF6, ZNF683, ZEB2, and PLZF; Fig 7C). We examined the temporal expression patterns of genes that were highly expressed in BATF3-OE cells and located in a relatively lower range of memory correlation scores (Fig S5A). We also analyzed the expression patterns of genes that were highly expressed in BATF-OE cells and located in a relatively higher range of memory correlation scores (Fig S5B). We examined the expression dynamics of genes previously reported to promote memory formation (Fig S5C). Transcription factors such as TCF7, PRDM1, and ZEB2, which play important roles in T-cell differentiation, were plotted in the memory correlation score closer to the effector side (Fig 7A). Their expression patterns increased on day 7, followed by a decrease on day 14. In contrast, transcription factors MAF, MXD4, and KLF6 showed the opposite pattern, being down-regulated on day 7 and up-regulated on day 14 (Fig 7B and C). To investigate the relevance of BATF3 and the genes identified in CD19 CAR-T cell therapy, we analyzed publicly available data. Using single-cell RNA-seq data from responders and nonresponders after tisa-genlecleucel treatment (43), we identified genes that were significantly differentially expressed between the two groups on day 7 of CAR-T cell infusion (Fig 7D). A secondary analysis was conducted using previously published RNA-seq data from CD8$^+$ T cells overexpressing BATF3 and GFP (44), to investigate the expression of the top 168 genes identified by the adjusted $P$-value ranking of comparisons between responders and nonresponders from tisagenlecleucel treatment. Of the 168 genes, 23 associated with responders and 8 with nonresponders were significantly regulated in BATF3-OE cells compared with GFP-OE CD8$^+$ T cells (Fig 7E). This set included KLF6, which was plotted in the BATF3-high expression region (Fig 7A). In a similar analysis using data from patients treated with axicabtagene ciloleucel, PRDM1 and ZEB2 were significantly up-regulated in the responder group (Fig S6), which differed from the data from the tisagenlecleucel (Fig 7D) (43). Collectively, BATF3 and BATF regulate distinct transcription factor networks, with BATF3-associated genes (including KLF6) enriched in responding CAR-T patients, linking BATF3-mediated memory programming to clinical therapeutic outcomes.

## Discussion

In the present study, we screened genes associated with a sustained IL-2 production capacity in human virus–specific CD8$^+$

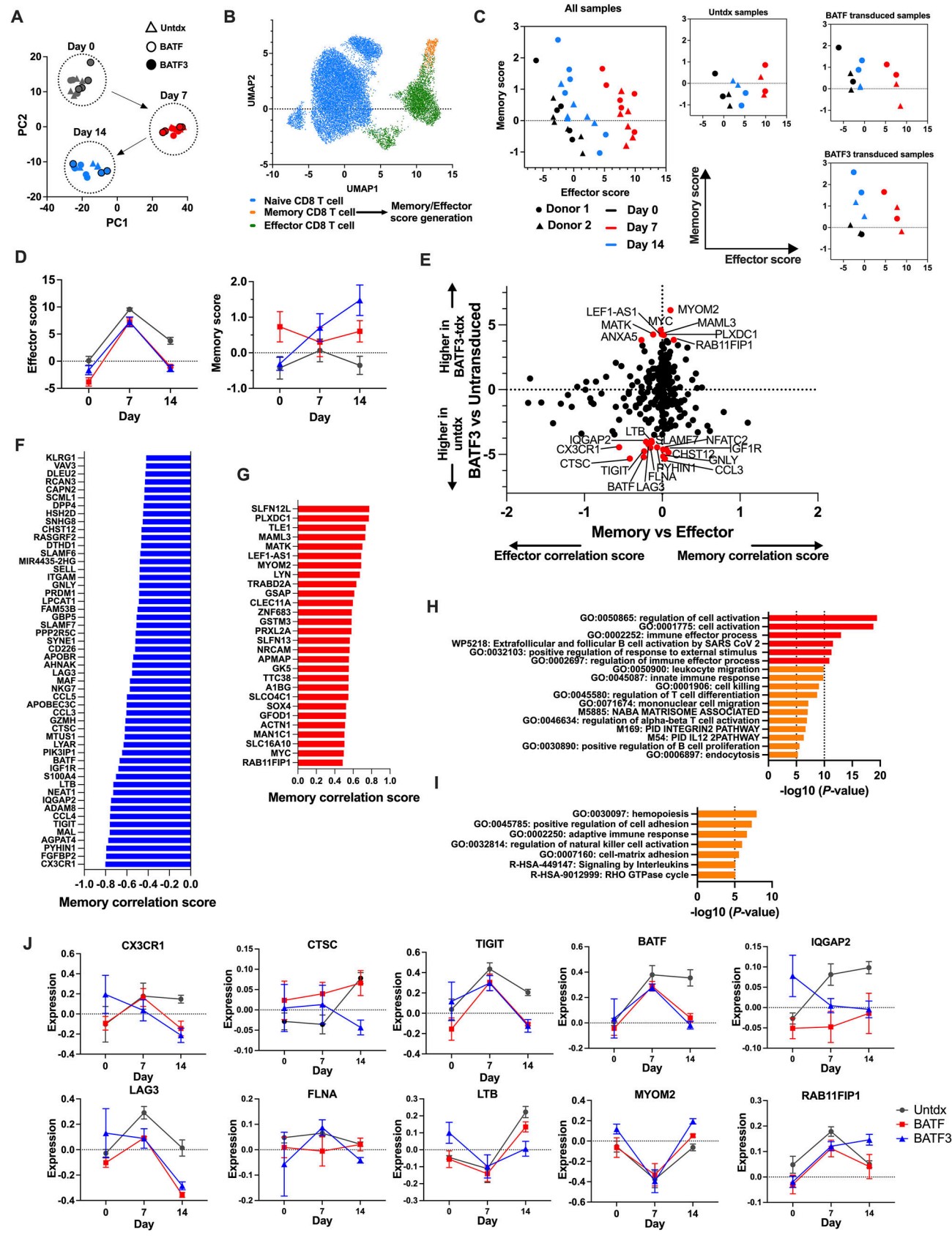

T cells and chose several TFs as candidates. Among these, introduction of *BATF3* improved CAR-T cell proliferation when overexpressed, and we further focused on the AP-1 TF family, including *c-JUN* and *BATF* (37, 38, 39). We transduced each of the AP-1 TF family genes and the CD19 CAR gene into CD8[+] T cells; then, CD19 CAR-T cells with *BATF3* showed significantly higher proliferation compared with CD19 CAR-T cells with other TFs. In contrast, the cytokine production of CD19 CAR-T cells expressing *BATF3* was lower than that of the other CD19 CAR-T cells.

We also confirmed that virus-specific T cells overexpressing BATF or BATF3 proliferated more efficiently. Because CMV-CTL showed lower expression of BATF and BATF3 than Flu-CTL at screening, the improvement in proliferative function by the overexpression of these genes was more marked in CMV-CTL than in Flu-CTL. We also found that CD19 CAR-T cells with BATF3 proliferated better than other CD19 CAR-T cells. Meanwhile, CD19 CAR-T cells with BATF proliferated like CD19 CAR-T cells with c-Jun and control vector. We initially focused on *BATF3* and other related genes within a cell population with higher IL-2 expression; however, ultimately, the correct description was that IL-2 expression is sustained for a longer period. The results of our experiments with CD19 CAR-T cells are consistent with the results of another study (33). In CD8[+] T cells, BATF works in differentiation to effector cells and proliferation of them (34, 35, 36). The functions of BATF are likely advantageous in the proliferation of virus-specific T cells, because naive T cells predominate at the initiation of the experiment. In experiments with virus-specific T cells and CD19 CAR-T cells, T cells expressing BATF3 continued to proliferate after repeated stimulation. This suggests that less differentiated cells were preserved after repeated stimulation.

BATF3-OE enhanced the antitumor effect of CAR-T cells in a previous report (44). This also suggests that both BATF and BATF3 work cooperatively in the effector phase. Meanwhile, BATF3 is an essential factor for the formation of memory cells (33). Therefore, we generated BATF-OE and BATF3-OE CAR-T cells and performed ATAC-seq using these cells. BATF-OE and BATF3-OE CAR-T cells showed similar kinetics during the effector phase. However, in terms of memory scores, there was a clear difference between days 7 and 14, with a significant increase observed in the BATF3-OE CAR-T cells. These findings suggest that these two TFs play different roles in the transition from the effector phase to the memory phase.

Whereas extended antigen-free culture would provide direct demonstration of memory characteristics, multiple lines of evidence support that BATF3-OE cells would show superior performance in such assays: progressive enrichment over 30–40 d indicates sustained survival capacity; memory-associated chromatin accessibility patterns are established and stable; clinical data demonstrate that BATF3-regulated programs correlate with durable in vivo responses requiring months of persistence; and published studies showed BATF3 promotes memory T-cell survival during antigen-free periods (33). The convergence of molecular, functional, and clinical evidence substantially supports our conclusions about BATF3's role in promoting memory-associated characteristics.

In CD8[+] T cells, BATF promotes T-cell exhaustion when antigen exposure continues (36). In our study, BATF-OE virus-specific T cells and CD19 CAR-T cells showed a decrease in the percentage of BATF-positive cells after repeated stimulation, making them less likely to proliferate. Because BATF3 is reported to act complementarily to BATF in some helper T cells (45), BATF3 may act similar to BATF. However, our data showed that BATF3-OE CD19 CAR-T cells tended to express lower levels of CTLA-4, PD-1, and TIM-3 than c-JUN-OE CD19 CAR-T cells after repeated stimulations. Therefore, BATF3 may function in cooperation with BATF to increase the number of effectors during the effector phase. BATF and BATF3-OE do not inhibit effector cell differentiation; rather, both TFs initially promote effector differentiation. Once the effector phase is complete, BATF3 may contribute to create memory cells. In *ALK*[+] anaplastic large cell lymphoma, classical Hodgkin Lymphoma, and adult T-cell leukemia/lymphoma, *BATF3* up-regulates *Myc* and plays an important role in tumor growth and survival (46, 47). Normal T cells act differently from tumor cells. However, BATF3-OE CD8[+] T cells may also be associated with cell survival and the *BATF3*-up-regulated *Myc* pathway.

From the results of ATAC-seq, we extracted genes associated with a low memory score and found that they included exhaustion markers, such as LAG3 and TIGIT (48, 49), as well as PRDM1, which increase memory cells when knocked out (50, 51). CX3CR1 was also included and is known as a marker of precursor exhausted cells (52). BATF itself was also found among these genes, which is an important observation. In contrast, the list of genes associated with a high memory score included genes involved in prosurvival signals, such as SOX4 and MYC (53). Among the genes associated with low and high memory scores, only a small number showed changes in contribution between BATF-OE and BATF3-OE from day 7 to day 14. Genes related to cell metabolism, such as *CTSC* and *RAB11FIP1*, were also identified. Cathepsin C plays a central role in the activation of many serine proteases in immune cells (54). RAB11FIP1 is a member of the Rab11 family of interacting proteins,

**Figure 6. BATF3 mediates memory T-cell formation through dynamic reorganization of a number of genes.**
**(A)** PCA of untransduced, BATF-transduced, and BATF3-transduced CD19 CAR-T cells. After TF transduction, as shown in Fig 5A, CD19 CAR-T cells were stimulated with γ-irradiated K562-CD19 cells. Before stimulation, on days 7 and 14, CD19 CAR-T cells were harvested and used for ATAC-seq. Samples were prepared from three different donors and duplicated for each condition. **(B)** Effector score and memory score generation. From public data, we extracted naïve, effector, and memory CD8[+] T cells and created an effector signature from the change from naïve to effector and a memory signature from the change from effector to memory. The effector and memory scores were calculated using the created signatures. **(C)** Alterations in the effector and memory scores from days 0 to 14. **(D)** Summary of alterations in effector and memory scores. **(E)** Scatter plots of the BATF3 group compared with the untransduced group on the vertical axis and the contribution to memory and effector scores on the horizontal axis. The upward direction indicates a higher contribution of each gene to the BATF3 group, and the rightward direction indicates a higher contribution of the gene to the memory score. **(F, G)** Bar graph plotting genes with low and high memory scores in the BATF3 group compared with the untransduced group. **(H, I)** Pathway analyses of low and high memory score genes in the BATF3 group compared with the untransduced group, respectively. **(J)** Expression of individual genes. Genes exhibiting a high contribution to the memory score were selected (seven low memory correlation score genes and three high memory correlation score genes). Data are presented as the mean ± SEM. Significance levels are shown only for comparisons with significance (one-way ANOVA on day 14, *P < 0.05, **P < 0.01).

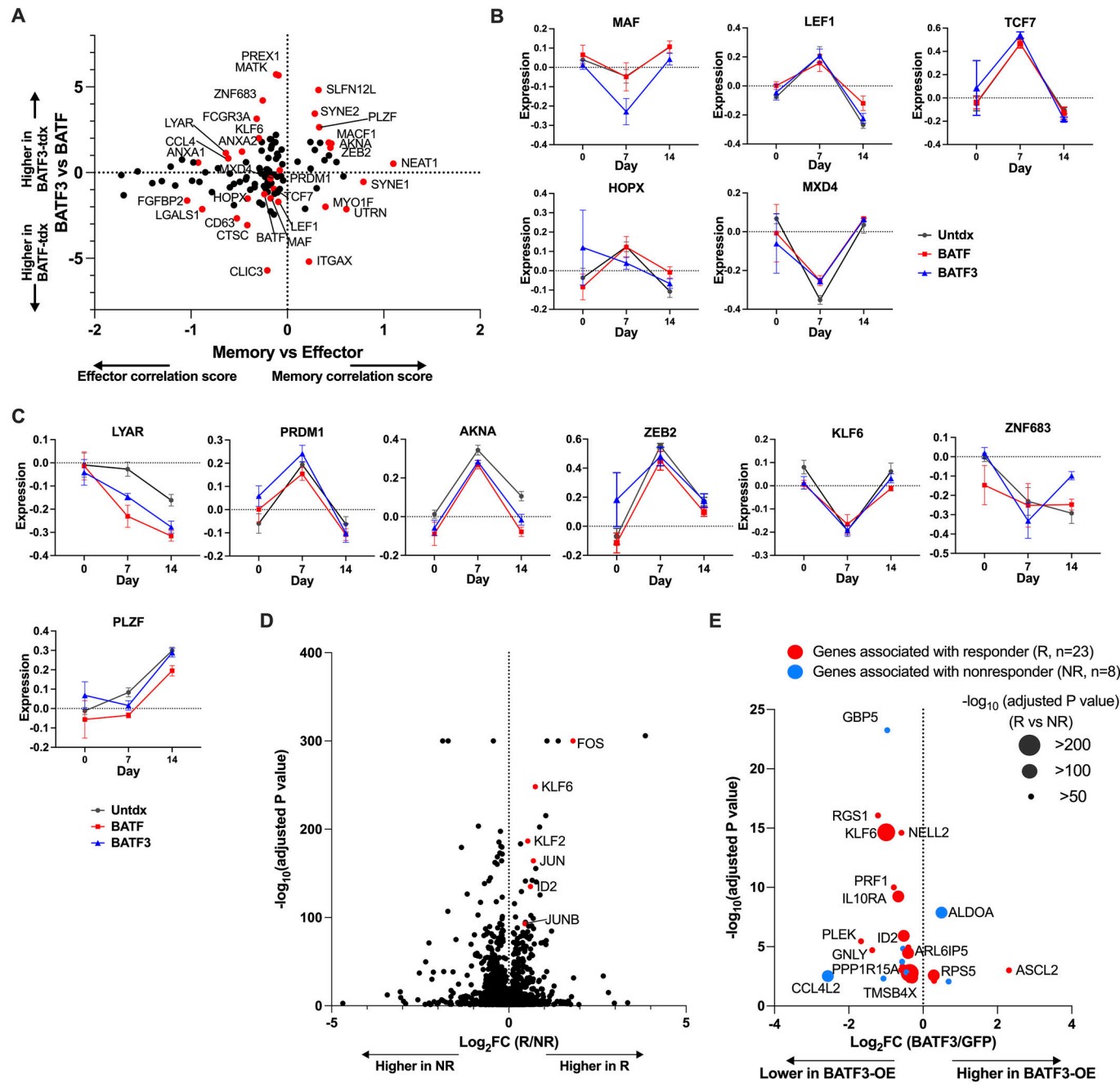

**Figure 7. Identification of relevant TFs by comparing BATF3-OE and BATF-OE CAR-T cells.**
**(A)** Scatter plots of the BATF3-OE group compared with the BATF-OE group on the vertical axis and the contribution to memory and effector scores on the horizontal axis. The upward direction indicated a higher contribution of each gene to the BATF3-OE group, and the rightward direction indicated a higher contribution of the gene to the memory score. **(B, C)** Expression of individual TF genes. **(B, C)** Genes exhibiting higher expression in the BATF-OE group (B) and the BATF3-OE group (C). **(D)** Reanalysis of single-cell RNA-seq data from patients treated with tisagenlecleucel. Volcano plot showing DEGs between responders and nonresponders. **(E)** Reanalysis of RNA-seq data from comparisons between BATF3-OE and GFP-OE CD8⁺ T cells. Significance versus fold change between BATF3-OE and control (GFP)-OE CD8⁺ T cells for 168 genes associated with the clinical outcome of CD19 CAR-T cell therapy. **(B, C)** Data are presented as the mean ± SEM in (B, C).

affects the Rab11-mediated recycling of vesicles, and is involved in endosomal trafficking (55). This suggests that memory cell formation may be promoted by altering cellular metabolism, including protein and vesicle recycling. Pathway analyses of genes with a high contribution to the effector score in BATF3-OE cells and those with a high contribution to the memory score showed similar top-ranking pathways. This suggests that opposite processes occur in related pathways, such as cytoskeletal regulation and metabolism, during the transition from the effector phase to the memory phase. Using an independent publicly available dataset of

PB samples from patients treated with tisagenlecleucel (43), we further evaluated the overlap between the top 168 genes differentially expressed between responders and nonresponders and those significantly regulated by BATF3-OE CD8+ T cells (44). This integrative analysis identified KLF6 and IL-10RA as genes enriched in the responder group and significantly regulated by BATF3-OE, suggesting a potential mechanistic link. In BATF3-OE CAR-T cells, KLF6 expression appeared to be lower than that in the control both at the prestimulation stage and on day 14 after stimulation. Meanwhile, it was slightly higher than that in the control on day 7. This pattern was consistent with the observation that KLF6 expression was elevated in the day 7 tisagenlecleucel dataset (Fig 7D), whereas it was down-regulated in the prestimulation BATF3/GFP-OE comparison (Fig 7E). The functional relevance of these genes in modulating CAR-T cell persistence and antitumor activity warrants further investigation.

The positioning of classical differentiation-associated transcription factors in our memory correlation score analysis requires clarification. Our memory score is a computational metric derived from single-cell transcriptome data comparing effector versus memory CD8+ T-cell states during a specific transition phase, rather than a universal definition of memory programs. Classical TFs like TCF7, PRDM1, and ZEB2 have dynamic, stage-specific roles (50, 51, 56): TCF7 is critical for establishing memory precursors during initial effector responses but may be transiently elevated at day 7 before decreasing by day 14 (Fig 7B and C), consistent with a transition role rather than memory maintenance; PRDM1 has dual roles in promoting effector function while preventing exhaustion, with optimal memory requiring its down-regulation after initial effector phase; and ZEB2 drives effector differentiation but can promote terminal differentiation when chronically elevated. Our signature specifically captures the day 7 to 14 transition under repetitive CAR-T stimulation, which represents memory maintenance under chronic antigen exposure rather than initial memory formation after acute infection. Conversely, factors positioned toward the memory side—MAF (opposes terminal differentiation), MXD4 (antagonizes Myc-driven effector programs), and KLF6 (our novel finding enriched in responding CAR-T patients, Fig 7D and E)—may be particularly important for memory maintenance under chronic stimulation conditions. Despite counterintuitive positioning of some classical factors, the score's biological relevance is validated by a series of our experimental data: specific increases only in BATF3-OE cells (Fig 6D); functional memory characteristics of BATF3-OE cells (survival, low exhaustion markers); clinical correlation with CAR-T response (Fig 7D and E). Thus, we believe that our score reflects a context-specific metric capturing memory programming relevant to CAR-T persistence under chronic stimulation, rather than a universal classification system.

We identified *BATF3* as a gene associated with less differentiated T lymphocytes by comparing flu-specific CTLs and CMV-specific CTLs. BATF3-OE rendered both virus-specific CTLs and CAR-T cells resistant to repeated stimulation. The proportion of BATF3-positive cells increased with each round of stimulation. Compared with other AP-1 TF family, BATF and BATF3 showed similar resistance to repeated stimulation. ATAC-seq analysis comparing BATF-OE and BATF3-OE cells showed that both BATF and BATF3 contribute to the induction of the effector phase. However,

BATF3 plays a larger role in the transition from the effector phase to the memory phase. However, the detailed molecular mechanism by which BATF3 promotes memory cell formation remains unclear. However, further studies are needed to clarify the mechanism that may help develop stronger lymphocytes against infections and tumors in the future.

# Materials and Methods

### Human subjects

The research protocols were approved by the Institutional Review Board of the Nagoya University Graduate School of Medicine (approval numbers: 2022-0093 and 2022-0265). PBMCs were obtained from healthy volunteers. Written informed consent was obtained from each donor, in accordance with the Declaration of Helsinki.

### Cell lines

K562 and K562 transduced with HLA-A*24:02 and B*07:02 or CD19 (K562-CD19) were maintained in our laboratory (57). We created K562 cells expressing HLA molecules (K562-HLA) by transducing K562 cells with HLA-A*02:01, A*24:02, or B*07:02. In addition, K562 cells were transduced with CD19 (K562-CD19). The authenticity of all the cell lines was routinely validated by examining their immunophenotypes using flow cytometry. Cells were cultured for a maximum of 2 mo before use. Cell lines were cultured in RPMI 1640 medium containing 10% fetal bovine serum, 0.8 mM L-glutamine, and 1% penicillin–streptomycin. K562 cells were retrovirally transduced to express truncated CD19 (K562-CD19) as described previously (57).

### Virus-specific CTL generation and evaluation

PBMCs from healthy donors were stimulated with virus-specific peptide (5 µM)-pulsed K562-HLA. For CMV, the peptides included pp65 (NLVPMVATV for HLA-A*02:01, QYDPVAALF for HLA-A*24:02, and TPRVTGGGAM for HLA-B*07:02). For influenza, the M1 peptide GILGFVFTL for HLA-A*24:02 was used in all donors. Recombinant human IL-2 (IL-2, 50 U/ml) was added on day 1 and subsequently every 3–4 d until day 10. On day 10, the cells were either restimulated with the same viral peptide-pulsed K562-HLA or analyzed by flow cytometry for antigen-specific CTL detection and cytokine production. For restimulation, K562-HLA cells were pulsed with 5 µM of the corresponding viral peptide at RT for 2 h and then irradiated (100 Gy) before co-culturing with PBMCs at a 1:1 ratio. Cultures were extended until day 30 with restimulation every 10 d.

### RNA extraction and RNA sequencing

mRNA was extracted from cells using RNA Blood Mini Kit (QIAGEN) according to the manufacturer's instructions. RNA-seq libraries were prepared using NEBNext Ultra II RNA Library Prep Kit for Illumina (New England Biolabs) and NEBNext Poly(A) mRNA

Magnetic Isolation Module (New England Biolabs) following the manufacturer's instructions and sequenced on an Illumina HiSeqX with 2 × 150 base paired-end reads. First, the raw read data quality was assessed for each sample using FastQC (version 0.11.9) for RNA-seq analysis. Transcript abundance was then directly quantified from the raw RNA-seq FASTQ files using the Kallisto v0.44.0 pseudoalignment method (58). For rapid and accurate quantification, a transcriptome index constructed from the Ensembl project's transcriptome v91 with a 100 bootstrap value based on the pseudoalignment was used. Gene-by-gene expression matrices were created using the R package tximport. Differential expression was analyzed using an integrated web application for differential expression and pathway analysis of RNA-seq data (iDEP) (59). For differential expression analysis, we used an adjusted $P$-value threshold of 0.1, which is appropriate for an exploratory screening phase given our sample size (n = 3 donors per group). This threshold balances the risk of false negatives (missing true candidate genes) against false positives, with the understanding that all candidates would be subsequently validated functionally. The use of $P < 0.1$ for discovery-phase transcriptomic screening with subsequent functional validation is an accepted approach in the field (59).

### Viral vector construction

CD19 CAR, which includes CD28 costimulatory domain, was generated by fusing the costimulatory domain with anti-CD19 single-chain variable fragment (scFv)-hinge-CD28 transmembrane domain followed by the CD3ζ intracellular domain (60, 61). The CD19-scFv was based on the clone FMC63 (62). CD19 CAR constructs were then fused with a truncated version of the epidermal growth factor receptor (tEGFR) that lacked epidermal growth factor binding and intracellular signaling domains downstream of the self-cleaving T2A sequence. tEGFR was used as a transduction and selection marker using biotinylated Erbitux mAb (Bristol Myers Squibb) (63, 64, 65). The CAR constructs were subcloned into the LZRS-pBMN-Z vector. Gamma-retroviral supernatants were obtained using a Phoenix-Ampho system (Orbigen).

To transduce TF, we prepared each TF packaged with EGFP/GFP, both as a retrovirus and as a lentivirus. For retrovirus and lentivirus generation, we used the LZRS-pBMN-Z vector (TF-IRES-EGFP) and a bidirectional promoter lentivirus vector (pCDH-EF1α-MCS-PGK-GFP, System Biosciences), respectively. The packaging and concentration methods for the lentiviruses have been described previously (60).

### Generation, expansion, and selection of CD19 CAR-T cells

PBMCs were obtained from healthy donors and isolated by whole-blood centrifugation using Ficoll-Paque (GE Healthcare). CD8[+] lymphocytes were purified using immunomagnetic beads (Miltenyi Biotec, Bergisch Gladbach) and activated using anti-CD3/CD28 beads (Invitrogen). Retroviral transduction was done on day 3 using recombinant human RetroNectin fragment-coated plates (Takara Bio) and centrifugation at 950$g$ for 1 h at 32°C. Transduced T cells were expanded in RPMI 1640 medium containing 10% human serum, 0.8 mM L-glutamine, 1% penicillin–streptomycin, and 0.5 $\mu$M

2-mercaptoethanol, which was supplemented with 50 IU/ml recombinant human IL-2. On day 7, the CAR-positive T cells were enriched with biotin-conjugated anti-EGFR mAb and streptavidin beads (Miltenyi Biotec). The enriched CAR-positive cells were subsequently expanded by culturing with CD3/28 beads at a 1:3 responder-to-bead ratio and supplemented with 50 IU/ml IL-2 on days 1, 4, and 7 (57). After 10 d, the expanded CAR-T cells were harvested and used for downstream experiments. Regarding T-cell phenotypes, the CAR-T cells were stimulated weekly with γ-irradiated K562-CD19 cells at a 1:1 ratio for three consecutive weeks. The tEGFR-transduced T cells were cultured in parallel according to the same procedure used for transduced T cells, except for the addition of the virus.

For AP-1 TF family transduction experiments, CD8[+] T cells were transduced on day 1 with lentiviral vectors encoding BATF3, BATF, c-JUN, or a control vector in the presence of polybrene (2 $\mu$g/ml) and IL-2 (50 IU/ml) by centrifugation at 950$g$ for 45 min at 32°C. These cells were subsequently transduced with the CD19 CAR retroviral vector on day 3.

For retroviral TF transduction, CD8[+] T cells were simultaneously transduced with both the TF and CD19 CAR retroviral vectors on day 3. The transduction was performed using plates coated with recombinant human RetroNectin fragment with centrifugation at 950$g$ for 2 h at 32°C.

For all experimental conditions, transduced T cells were expanded in CTL medium supplemented with IL-2 (50 IU/ml). On day 7, CAR-positive T cells were enriched using a biotin-conjugated anti-EGFR antibody and streptavidin beads (Miltenyi Biotec). Cells were cryopreserved in liquid nitrogen on day 11.

### CAR-T cell proliferation assay

For expansion assays, cryopreserved CAR-T cells were thawed and cultured in CTL medium. On day 0, the CAR-positive T cells were mixed with γ-irradiated (75 Gy) EBV-LCLs at a 1:7 ratio. IL-2 (50 IU/ml) was added on days 1, 4, and 7 (57). The restimulation cycles were repeated three times. After each cycle, the cells were analyzed for their phenotype and cytokine production. To assess proliferation capacity, both untransduced T cells and CD19 CAR-T cells were stimulated with γ-irradiated CD19-K562 cells at a 1:1 ratio. T-cell expansion was quantified by counting viable cells at multiple time points: after 72 h and continuing for up to 2 wk. Cell counts were performed using the trypan blue exclusion assay to determine cell viability.

### Immunophenotyping

All the samples were analyzed using flow cytometry on a FACSAria II flow cytometer (BD Biosciences), and the data were analyzed using FlowJo software (Tree Star). Human T cells were stained with combinations of the following mAb conjugated to fluorophores: CD3; CD8; CD45RA; CD62L; IL-2; IFN-γ; PD-1; LAG3 (BD Biosciences); mCD45; CTLA-4; and TIM-3 (BioLegend). Biotinylated Erbitux and streptavidin–phycoerythrin (BD Biosciences) were used to identify tEGFR[+] cells. Antigen-specific CTLs were detected using a CD8-APC antibody (BD Biosciences) and virus peptide-specific TCR tetramers (PE, MBL Life Science). For CAR-T cell phenotyping, the cells

were stained with CD8-APC, assessed for GFP expression, and analyzed for memory subsets using CD62L-PE-CF594 and CD45RA-PE-Cy7 antibodies (BD Biosciences). All antibodies used for flow cytometry analysis were applied at a concentration of 1 ng/μl or diluted at a ratio of 1:50.

## Intracellular cytokine staining and cytokine secretion assay

Untransduced or CD19 CAR-T cells were plated at an E:T ratio of 1: 2 with either K562 or CD19-K562 cells in the presence of brefeldin A (Sigma-Aldrich) for 4 h. The stimulated cells were then fixed and permeabilized with Cell Fixation/Permeabilization Kit (BD Biosciences) and stained with anti-IFN-γ or IL-2 mAb to detect intracellular cytokines. For the cytokine secretion assay, untransduced or CD19 CAR-T cells were co-cultured at a 1:1 ratio with γ-irradiated CD19-K562 cells for 16 h. Supernatants from co-cultures were collected, and the IL-2, IFN-γ, and TNF-α concentration was measured with an ELISA (BD Biosciences).

## qRT-PCR

RNA was extracted and purified from the cells using QIAamp RNA Blood Mini Kit (QIAGEN) according to the manufacturer's instructions. Reverse transcription was performed by inputting an equal mass of mRNA for each sample into a 10 μl SuperScript Vilo cDNA Synthesis reaction (Invitrogen). qRT-PCR was performed to evaluate the expression of BATF3, BATF, JUN, and FOSL1 using HPRT as a housekeeping gene. Predesigned PrimeTime qRT-PCR assays were used for BATF3 (Assay ID: Hs.PT.58.2603224), BATF (Hs.PT.58.2200443), and JUN (Hs.PT.58.25094714), FOSL1 (Hs.PT.58.2855727), and HPRT (Hs.PT.58v.45621572) (all from Integrated DNA Technologies). The synthesized cDNA and primers were mixed with PrimeTime Gene Expression Master Mix (Integrated DNA Technologies), and qRT-PCR was performed using Applied Biosystems 7300 Real-Time PCR System (Thermo Fisher Scientific). The reaction mixture was dispensed into MicroAmp Optical 96-Well Reaction Plate (Thermo Fisher Scientific). The PCR cycling protocol was as follows: polymerase activation: 95°C for 3 min (1 cycle); denaturation: 95°C for 15 s; annealing/extension: 60°C for 1 min (45 cycles); and final cooling: 4°C. Relative quantification of qRT-PCR results was performed using the ΔΔCt method with HPRT as the control.

## ATAC-seq and data analyses

### Data generation
CD19 CAR-T cells were generated from two healthy volunteer donors, as previously described. After purification of CD19 CAR-T cells using tEGFR, the CAR-T cells were stimulated with K562-CD19 cells, and the CAR-T cells were harvested before stimulation on days 7 and 14. The samples were prepared in duplicate. ATAC-seq was performed using the Active Motif ATAC-seq kit (Cat. #53150), following the manufacturer's protocol. Briefly, the cells were collected, washed with PBS, and lysed to isolate the nuclei. The transposase reaction was performed by incubating the nuclei with Tn5 transposase, which fragmented the DNA and inserted sequencing adapters. After purification, the transposase-treated DNA fragments were amplified by PCR. The libraries were then

quantified, quality-checked using a bioanalyzer, and sequenced on an Illumina NovaSeq X Plus in paired read mode with a read length of 150 nucleotides (Macrogen). Data were processed and analyzed using standard ATAC-seq bioinformatics pipelines.

### Data processing
ATAC-seq data were analyzed using the nf-core ATAC-seq pipeline (41). Raw sequencing reads were first assessed for quality using FastQC (v0.11.9) and trimmed with Trim Galore (v0.6.7) to remove adapter sequences and low-quality bases. The trimmed reads were then aligned to the reference genome (GRCh38) using Bowtie2 (v2.4.5) with the default parameters. After alignment, duplicate reads were marked and removed using Picard (v2.27.4), and the mitochondrial reads were filtered to retain nuclear chromatin accessibility signals. Peak calling was performed using MACS2 (v2.2.7.1) with a Q-value cutoff of 0.05 to identify accessible chromatin regions. The number of reads within each peak was quantified for each sample using featureCounts software (v2.0.1) to generate a peak-by-sample read-count matrix. The raw counts were first normalized by dividing by peak length, followed by scaling based on the mean across samples. Peaks with low counts were excluded from further analysis. Subsequently, library size normalization and log transformation were performed. Highly variable peaks were identified, and principal component analysis (PCA) was conducted to visualize the distribution of samples according to subject ID and sampling day.

### DEG analysis using publicly available dataset
Single-cell RNA-sequencing (scRNA-seq) data from day 7 PB post-CD19 CAR-T cell infusion (43) were obtained from the Gene Expression Omnibus under the accession number GSE197268. The raw count matrices were processed in Python (version 3.11.7) using the Scanpy library (version 1.11.3) (66). After quality control, cells corresponding to the CD8$^+$ T-cell cluster were isolated for downstream analyses. DEG analysis between the responder and non-responder groups within this CD8$^+$ T-cell population was performed using the nonparametric Wilcoxon rank-sum test, as implemented in the rank_genes_ group function. This test was applied to the log-normalized expression values. Genes were selected based on the criteria of fold change ≥ 2 and ranking within the top 100 by test statistic. These selected genes were subsequently used for the calculation of memory and effector scores.

## Calculation of memory and effector scores

The correspondence between ATAC-seq peaks and genes was defined based on whether a peak overlapped any part of the gene body. The expression count of each gene was defined as the average of counts across all peaks assigned to that gene. A gene expression matrix was then constructed, and for each sample, the Δ expression was calculated by subtracting the mean expression at day 0. Average expression vectors were computed for memory, effector, and naïve CD8 T cells in the reference dataset. The scores were defined as follows:

Effector score: Calculated as the inner product between the Δ expression vector of each sample and the difference vector of effector versus naïve cells.

Memory score: Calculated as the inner product between the Δ expression vector of each sample and the difference vector of memory versus effector cells.

This approach allowed for quantitative assessment of the bias toward effector and memory lineages for each sample.

## Statistical analysis

All the experimental data are presented as the mean ± SEM or SD. The differences among the results were evaluated using *t* test and one-way ANOVA with Bonferroni's or Tukey's posttest correction when appropriate. Differences were considered statistically significant at $P < 0.05$. Statistical analyses were performed using GraphPad Prism version 9.1.2 software (GraphPad Software).

# Data Availability

All data associated with this study are presented in the article or Supplementary Information files. All RNA-seq data have been deposited with links to BioProject accession number PRJDB40675 in the DDBJ BioProject database. All ATAC-seq data were deposited in the Sequence Read Archive under the GEO repository accession no. GSE325835.

# Supplementary Information

# Acknowledgements

The authors thank the Division of Experimental Animals and the Division of Medical Research Engineering, Nagoya University Graduate School of Medicine, for their technical assistance.

## Author Contributions

K Umemura: data curation, formal analysis, investigation, and writing—original draft.
Y Kojima: data curation, software, formal analysis, investigation, and writing—original draft, review, and editing.
J Julamanee: conceptualization, data curation, formal analysis, investigation, and writing—original draft, review, and editing.
Y Okuno: formal analysis and investigation.
Y Takeuchi: investigation.
F Ohara: investigation.
S Kuwano: data curation, formal analysis, investigation, and writing—original draft.
Y Adachi: investigation.
R Hanajiri: investigation.
S Terakura: conceptualization, data curation, formal analysis, supervision, funding acquisition, investigation, methodology, project administration, and writing—original draft, review, and editing.

H Kiyoi: conceptualization, supervision, funding acquisition, and writing—review and editing.

## Conflict of Interest Statement

H Kiyoi has received research funding from Chugai Pharmaceutical Co., Ltd., Kyowa Hakko Kirin Co., Ltd., Zenyaku Kogyo Co., Ltd., FUJIFILM Corporation, Daiichi Sankyo Co., Ltd., Astellas Pharma Inc., Otsuka Pharmaceutical Co., Ltd., Nippon Shinyaku Co., Ltd., Eisai Co., Ltd., Pfizer Japan Inc., Takeda Pharmaceutical Co., Ltd., Novartis Pharma K.K., Sumitomo Dainippon Pharma Co., Ltd., Sanofi K.K., and Celgene Corporation; consulting fees from Astellas Pharma Inc., Amgen Astellas BioPharma K.K., and Daiichi Sankyo Co., Ltd.; and honoraria from Bristol Myers Squibb, Astellas Pharma Inc., and Novartis Pharma K.K.

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
