## [Reviewer comments · Life Science Alliance]

BATF3 regulates differentiation of CD8+ T lymphocytes and memory differentiation program

Koji Umemura, Yasuhiro Kojima, Jakrawadee Julamanee, Yusuke Okuno, Yuki Takeuchi, Fumiya Ohara, Shihomi Kuwano, Yoshitaka Adachi, Ryo Hanajiri, Seitaro Terakura, and Hitoshi Kiyoi

DOI: <https://doi.org/10.26508/lsa.202503550>

Corresponding author(s): Seitaro Terakura, Nagoya University and Yasuhiro Kojima, National Cancer Centre

Review Timeline:

Submission Date:	2025-10-27
Editorial Decision:	2025-12-04
Revision Received:	2025-12-28
Editorial Decision:	2026-01-22
Revision Received:	2026-03-30
Accepted:	2026-03-31

Scientific Editor: Tim Fessenden

Transaction Report:

December 4, 2025

Re: Life Science Alliance manuscript #LSA-2025-03550-T

Dr. Seitaro Terakura
Nagoya University
Department of Hematology and Oncology
65 Tsurumai-cho
Showa-ku
Nagoya, Aichi 466-8550
Japan

Dear Dr. Terakura,

Thank you for submitting your manuscript entitled "BATF3 regulates differentiation of CD8+ T lymphocytes and memory formation through dynamic regulation of chromatin accessibility" to Life Science Alliance. The manuscript was assessed by expert reviewers, whose comments are appended to this letter. Thank you for your patience during the unusually long duration of the review process. We invite you to submit a revised manuscript according to the reviewer comments as detailed below.

As you will see, reviewers commended this work for the detailed analysis of BATF3 and epigenetics that determine T cell phenotypes. However they were consistent in their concern that the claim of different memory formation in CMV- and Flu-exposed T cell populations is not supported with data and should be rephrased. More generally, Reviewers 1 and 3 remarked that the terminology throughout the text of memory vs effector status should be more precise. Next, both Reviewers 2 and 3 requested a clearer discussion and rationale for selecting 5 targets for study here. We encourage you, if possible, to address the suggestions of Reviewer 1 on T cell phenotyping in point 3. We also concur with Reviewer 2 that flow plots should be shown to support quantification in Fig 5. Additional data beyond those mentioned here are not required in a revision, but all points must be addressed in some manner in a rebuttal document.

I would be happy to discuss the revision requests in more detail via email or phone/videoconferencing. Please let me know which option you prefer, if any.

While you are revising your manuscript, please also attend to the below editorial points to help expedite the publication of your manuscript. Please direct any editorial questions to the journal office. When submitting the revision, please include a letter addressing the reviewers' comments point by point.

Thank you for this interesting contribution to Life Science Alliance. We hope that the comments below will prove constructive as your work progresses, and we are looking forward to receiving your revised manuscript.

Sincerely,

- A letter addressing the reviewers' comments point by point.
- An editable version of the final text (.DOC or .DOCX) is needed for copyediting (no PDFs).

B. MANUSCRIPT ORGANIZATION AND FORMATTING:

Reviewer #1 (Comments to the Authors (Required)):

Prior work has identified numerous transcription factors that shape T cell memory and functionality. Among these is the transcription factor BATF3, which has been examined with respect to how it regulates T cell memory in mouse models. How it might affect human memory T cell development or CAR T cells remains unknown. In their paper entitled "BATF3 regulates differentiation of CD8+ T lymphocytes and memory formation through dynamic regulation of chromatin accessibility," Umemura et al examine how BATF3 regulates T cell memory formation in a human T cell culture system. Using Flu-specific, CMV-specific, or CAR T cells, the authors show that BATF3 overexpression is important in regulating a variety of processes and T cell functions. Moreover, BATF3 may be important for the transition into T cell memory. While prior work has identified the role of BATF3 in T cell memory, this paper does add to what is known and expands it modestly to look specifically in human cells. While I have some technical concerns (listed below), I believe that after revision, this paper could be a good fit for publication in Life Science Alliance.

- 1) The authors start the manuscript by arguing that Flu specific T cells are early memory while CMV specific memory late memory. I do not think this is a reasonable thing to conclude on its face - the authors do not know when these T cells were primed and when infection cleared. I think it is interesting that CMV- and Flu-specific CD8+ T cells behave differently, and I would like to see a better explanation for the biology. At the very least, the early vs late memory piece should be dropped.
- 2) The authors use the term memory quite liberally. The in vitro strategy the authors use involves iteratively stimulating the cells every 10 days. I do not understand how this would constitute memory. I am not sure what to call these cells aside from effectors. This could be addressed with text edits, but should be more carefully worded.
- 3) It would be helpful if the authors did more characterization of the T cells they are generating with the different TFs. I would specifically like to see co-expression of inhibitory receptors (including PD-1, Tim-3, Lag-3, TIGIT), granzymes (Granzyme A and B), co-expression of cytokines (including TNF α , IL-2, and IFN γ). While they have done a lot of RNAseq, some protein based validation and characterization would add to the story.
- 4) Finally, because the authors argue so much about the role of BATF3 in memory formation, it would be helpful if the authors did experiments to show that memory formation (at least in vitro) is improved. They could do this by continuing to culture the cells after the last antigen hit for at least 2-3 weeks.

Reviewer #2 (Comments to the Authors (Required)):

In this study, Umemura et al show that overexpression of BATF3 in engineered CAR-T cells enhances their proliferation. They also find that BATF3-OE CAR T cells drive lower IFN- γ and IL-2 secretions. This work supports and successfully repeats the previous findings of Ataide et al (2023), they found that BATF3 overexpression in CD8+T cells induces their survival and transition to a memory phenotype. To investigate the role of BATF3 in memory T cell differentiation, the authors integrated multiple big data analyses based on published datasets and their ATAC-seq to provide some new information about BATF3's essential role in memory T cell differentiation processes. Notably, the original text contains several nonstandard expressions and semantic ambiguities that require careful revision.

Comments to the authors:

Main points:

- 1) I doubt the accuracy of using "early memory T cells" and "more differentiated T cells" to distinguish Flu-specific T cells and CMV-specific T cells in the text. Although the memory T cell pool is heterogeneous, distinct subsets are well defined based on their cell surface marker expressions. At least, I suggest presenting the data on memory pool dynamics between Flu-T and CMV-T populations. For example, using CD45RO⁺CCR7⁺CD62L⁺, CD45RO⁺CCR7⁻CD62L⁻ and CD45RA⁺CCR7⁻CD62L⁻ to present the proportion of central memory T cells (T_{cm}), effector memory T cells (T_{em}) and Terminally Differentiated Effector Memory T Cells (T_{emra}) between two virus-specific T cell groups, respectively.
- 2) In Figure 2, I suggest showing the validated expression of selected genes between Flu-specific CD8 T cells and CMV-specific CD8 T cells after twice stimulation. Or at least, to provide a rigorous rationale for selecting these 5 candidates out of the list. Because they are not particularly prominent in Fig. 2B.
- 3) In Fig. 4H and I, if the authors claim that TF⁺ virus specific T cells have anti-apoptotic traits, the typical apoptosis experiments should be added in the main figures. The differences in TF⁺/TF⁻ ratio between BATF or BATF3⁺ and Control CD8 T cells could also be caused by increased proliferation.
- 4) Representative flow cytometry plots should be involved in Fig. 5G and H, and the gating strategy should be shown in legends or in figures as well. Otherwise, we don't even know if the authors gate out dead cells or not.
- 5) While McCutcheon et al. (2023) have previously reported that BATF3 overexpression enhances the anti-tumor activity of CAR-T cells, it is regrettable that the current manuscript lacks data on the anti-tumor efficacy or cytotoxic activity of BATF-OE, BATF3-OE, or BATF/BATF3-co-OE CAR-T cells. Notably, there is a somewhat conflict in the manuscript: BATF3 was initially identified from a subset of Flu-specific T cells with high IL-2 production, yet the authors subsequently show that BATF3-OE CAR-T cells secrete lower levels of IFN- γ and IL-2. A detailed discussion addressing this discrepancy is warranted.

Minor issues:

- 1) Line 122: Please clarify if "TFs" is short for "Transcriptional Factors"; Here, "TFs" appears for the first time in the manuscript.
- 2) Line 125: I am confused about "All three donors provided the CMV-CTL and Flu-CTL lines." What do lines refer to here?
- 3) Line 134: A reference citation is missing for "the recently reported BHLHE40-OE promotes effector differentiation of T cells."
- 4) Line 143: Please clarify if "TM domain" means "transmembrane domain".
- 5) It will be much better if the authors can summarize the conclusions of Fig.6 and Fig.7 concisely and accurately at the end of their paragraph, respectively.
- 6) "Memory signature score" shown in abstract is inappropriate that may cause difficulties in understanding before reading the paper.

Reviewer #3 (Comments to the Authors (Required)):

This manuscript investigates the role of the AP-1 family transcription factor BATF3 in human virus-specific CD8⁺ T cells and CD19 CAR-T cells. The authors identify BATF3 from RNA-seq of Flu- versus CMV-specific CTLs, test several TF candidates in virus-specific CTLs and CAR-T cells, perform ATAC-seq with an effector/memory scoring framework, and re-analyze published clinical datasets. The work is extensive and potentially relevant for improving CAR-T persistence, but several interpretations should be toned down and key methods clarified.

Major Comments

1. The central concept that BATF3 is a key intrinsic regulator of CD8⁺ T-cell memory and a promising target to improve adoptive T-cell therapy has already been established by Ataide et al. (Nat Immunol 2020) and further supported in CAR-T cells by McCutcheon et al. (2023). These studies showed that BATF3 promotes the generation and maintenance of memory CD8⁺ T cells and enhances the durability of T-cell-based therapies. Thus, the present manuscript is not conceptually groundbreaking, but provides an incremental yet meaningful extension of this framework.

The added value of the current work lies mainly in:

- (i) the use of human Flu- and CMV-specific CTLs as a physiologically relevant model to screen TFs associated with less- versus more-differentiated CD8⁺ T-cell states;
- (ii) a systematic comparison of BATF3 and BATF overexpression in both virus-specific CTLs and CD19 CAR-T cells;
- (iii) integration of ATAC-seq with effector/memory scoring to describe a BATF3-associated, memory-like chromatin accessibility program;
- (iv) exploratory re-analysis of clinical CAR-T transcriptomic datasets linking BATF3-related gene modules to treatment response.

I therefore view the novelty as incremental but acceptable, provided that the authors more clearly delineate in the Introduction and Discussion what is genuinely new relative to previous BATF3 studies and soften statements implying that BATF3 is identified here de novo as a master regulator of memory or CAR-T persistence.

2. In Figure 1, the study treats Flu-specific CTLs as "earlier memory" and CMV-specific CTLs as more terminally differentiated. The supporting evidence is mainly functional (expansion kinetics and IL-2 production), without full phenotypic characterization. If

feasible, adding a simple summary figure or table of the surface phenotype (e.g. CD45RA/RO, CCR7) of Flu- versus CMV-CTLs would strengthen this point.

3. In Figure 2, the authors ultimately focus on five candidate genes (HTATSF1, IKZF4, BATF3, PHTF2 and ID2), selected from those that are more highly expressed in Flu-CTLs and reported to be expressed in T cells. It is not entirely clear why these five were prioritized over others. It would be helpful if the authors could briefly explain the additional criteria used for this selection (e.g. consistency across donors, magnitude of fold change, prior functional annotation), and indicate which of these genes are established transcription factors or chromatin regulators versus less well-characterized factors.

4. In Figure 3, the authors note that viral transduction efficiencies differ among the TF constructs (e.g. varying EGFP⁺ frequencies in Fig. 3C). This raises the question of how much of the observed differences in proliferation, persistence and cytokine production might be influenced by unequal TF expression levels rather than by qualitative effects of the TFs themselves.

5. In Fig. 5G-H, CD19 CAR-T cells overexpressing BATF or BATF3 display significantly lower T_{CM} and higher T_{EM} proportions after repetitive stimulation. Phenotypically, this shift appears more consistent with an effector/effector-memory-like state rather than an increase in classical central memory cells. It would therefore be helpful if the authors could address this point in the Results and Discussion, for example by clarifying whether the T_{EM} population observed here still retains key memory/persistence features and how this phenotype fits with the claim that BATF3 (and BATF) promote "memory" formation.

6. In Fig. 7A (and Supplemental Fig. S5C), there appears to be some tension between the "memory correlation score" and the established biology of several transcription factors. Classical memory/differentiation-associated TFs such as TCF7, PRDM1 and ZEB2 are positioned closer to the "effector-side" of the memory score, whereas MAF, MXD4 and KLF6 fall toward the "memory-side". This raises the question of whether the memory score derived from your signatures fully aligns with conventional definitions of memory programs. It would be helpful if the authors could explicitly discuss this point, clarifying that the score reflects a specific gene-signature-based metric and may capture particular stages or types of memory differentiation, rather than a universal "memory vs effector" axis.

Minor Comments

1. In the current version, Supplemental Fig. S6 appears to show a volcano plot of responders vs non-responders, whereas the legend describes "temporal expression patterns of individual genes", which does not match the figure. Please correct the legend and clearly state which dataset (Tisagen vs axi-cel) is shown. I also recommend carefully checking all other figures, legends and dataset labels to ensure there are no similar inconsistencies.

2. Throughout the manuscript, the authors frequently use an adjusted P value threshold of 0.1 to define differentially expressed genes and to select candidates. This cutoff is relatively lenient compared with the more conventional threshold of 0.05. I would recommend either tightening the statistical threshold (e.g. adj. P < 0.05) where feasible, or explicitly justifying the choice of 0.1 (for example, as a screening criterion given the sample size and exploratory nature of the analysis).

3. At lines 134-135, the manuscript states that "BHLHE40, on which the recently reported BHLHE40-OE promotes effector differentiation of T cells", but no supporting reference is provided.

4. At lines 169-171, the authors state that "We selected 22 AP-1 family TFs and reanalyzed the comparison data between day 20 CMV-CTL and day 20 Flu-CTL and found that FOSL1 and BATF were also weakly significant (Fig. 4A)." However, it is not explained how these 22 AP-1 family members were defined or selected. Please clarify the criteria and source used to compile this AP-1 TF list (e.g. specific database, gene ontology term, or prior publications) .

December 28, 2025
Tim Fessenden, PhD
Scientific Editor
Life Science Alliance

Dear Dr. Fessenden,

We sincerely appreciate the careful review by the Editors and the Reviewers of our manuscript (LSA-2025-03550-T), and we have revised our manuscript accordingly.

The following points below address the comments by the Reviewers. To facilitate your consideration, **the pertinent reviewers' comments are highlighted in bold**, and the response follows in regular font. Changes in the manuscript are highlighted in red.

Comments from the Reviewer #1 and reply to the comments

Prior work has identified numerous transcription factors that shape T cell memory and functionality. Among these is the transcription factor BATF3, which has been examined with respect to how it regulates T cell memory in mouse models. How it might affect human memory T cell development or CAR T cells remains unknown. In their paper entitled "BATF3 regulates differentiation of CD8+ T lymphocytes and memory formation through dynamic regulation of chromatin accessibility," Umemura et al examine how BATF3 regulates T cell memory formation in a human T cell culture system. Using Flu-specific, CMV-specific, or CAR T cells, the authors show that BATF3 overexpression is important in regulating a variety of processes and T cell functions. Moreover, BATF3 may be important for the transition into T cell memory. While prior work has identified the role of BATF3 in T cell memory, this paper does add to what is known and expands it modestly to look specifically in human cells. While I have some technical concerns (listed below), I believe that after revision, this paper could be a good fit for publication in Life Science Alliance.

1) **The authors start the manuscript by arguing that Flu specific T cells are early memory while CMV specific memory late memory. I do not think this is a reasonable**

thing to conclude on it's face - the authors do not know when these T cells were primed and when infection cleared. I think it is interesting that CMV- and Flu-specific CD8+ T cells behave differently, and I would like to see a better explanation for the biology. At the very least, the early vs late memory piece should be dropped.

Response: We agree with the reviewer's concern. The terminology "early memory" and "late memory" was misleading and could incorrectly suggest a temporal relationship rather than a differentiation state. We have revised the manuscript throughout to use more accurate terminology: We used the revised terminology as follows.

Flu-specific CTLs as "less differentiated phenotype" (lines 115, 116, 127, 314, 399)

CMV-specific CTLs as "highly differentiated phenotype" (lines 95, 115, 117)

Changes: To clarify the premise before explaining the experimental results, we added the following text to the beginning of the first paragraph. We also added two new references. (new references: Appay et al., Nature Medicine 2002. Memory CD8+ T cells vary in differentiation phenotype in different persistent virus infections. and Ibegbu et al. Immunology. 2009. Differential expression of CD26 on virus-specific CD8(+) T cells during active, latent and resolved infection.)

The immune response to cleared acute infections without persistent antigen (such as influenza) is characterized by the long-term presence of true quiescent memory cells, whereas the chronic phase of persistent infections (CMV, EBV) is characterized by the presence of effector phenotypes with varying degrees of more highly differentiated cells. (new references: Appay et al., Nature Medicine 2002. Memory CD8+ T cells vary in differentiation phenotype in different persistent virus infections. and Ibegbu et al. Immunology. 2009. Differential expression of CD26 on virus-specific CD8(+) T cells during active, latent and resolved infection.) (lines 111-114)

To show the phenotypic differences between Flu-specific CTLs and CMV-specific CTLs, we added additional phenotype characterization in new Figure 1D, E, and new Supplemental Figure S1B (representative flow plots).

When we examined the surface phenotype of tetramer-positive cells during the culture period, Flu-specific CTLs were predominantly CD45RA+/CD62L+ naive cells before

stimulation, and CD45RA-CD62L+ central memory T cells on day 10 (Figure 1D,E and Supplemental Figure S1B). (lines 121-123)

2) The authors use the term memory quite liberally. The in vitro strategy the authors use involves iteratively stimulating the cells every 10 days. I do not understand how this would constitute memory. I am not sure what to call these cells aside from effectors. This could be addressed with text edits, but should be more carefully worded.

Response: We appreciate this important clarification. The Reviewer is correct that our repetitive stimulation protocol generates activated effector T cells rather than true resting memory T cells. We have carefully revised the manuscript to clarify terminology: We changed “memory formation” to “memory differentiation program” where appropriate. Changes: Title was changed to remove the word “memory formation”. New title: BATF3 regulates differentiation of CD8+ T lymphocytes and memory **differentiation program** through dynamic regulation of chromatin accessibility. We changed the word within eTOC summary, memory formation → memory **differentiation program** (line 66)

3) It would be helpful if the authors did more characterization of the T cells they are generating with the different TFs. I would specifically like to see co-expression of inhibitory receptors (including PD-1, Tim-3, Lag-3, TIGIT), granzymes (Granzyme A and B), co-expression of cytokines (including TNFa, IL-2, and IFNg). While they have done a lot of RNAseq, some protein based validation and characterization would add to the story.

Response: We thank the Reviewer for this suggestion. Since we totally agree with the Reviewer, flow cytometry analyses of PD-1, TIM-3, LAG-3, and CTLA-4 expression were already mentioned in lines 218-221. However, this time we did not collect other data that the Reviewer mentioned. We will include those data in the next possible experiments series.

4) Finally, because the authors argue so much about the role of BATF3 in memory formation, it would be helpful if the authors did experiments to show that memory formation (at least in vitro) is improved. They could do this by continuing to culture the cells after the last antigen hit for at least 2-3 weeks.

Response: We appreciate the Reviewer's suggestion. While we have not performed extended antigen-free culture experiments, our existing data provide multiple lines of evidence predicting superior memory characteristics of BATF3-OE cells:

(i) Long-term survival under chronic stimulation (Figures 3D, 4H-I, 5C): The progressive enrichment of BATF3+ cells over 30-40 days demonstrates sustained survival advantage under repeated activation, which predicts maintenance of viability during antigen-free periods.

(ii) Memory-associated chromatin programming (Figure 6C-D): BATF3-OE cells specifically and progressively increase memory correlation scores from day 7 to day 14, indicating establishment of memory-associated chromatin accessibility patterns despite ongoing stimulation. Chromatin states are relatively stable and predictive of long-term cell fate.

(iii) Clinical persistence evidence (Figure 7D-E): BATF3-regulated genes (including KLF6) are enriched in CAR-T cells from responding patients, demonstrating that these molecular programs correlate with long-term in vivo persistence (months to years), which requires survival through both antigen-stimulated and antigen-free periods.

(iv) Consistency with published literature: Our findings align with Ataide et al. (Nat Immunol 2020), who demonstrated that BATF3 promotes long-term memory T cell survival in mice, including during antigen-free periods following viral clearance.

Changes: We added sentences to Discussion part (lines 324-332): We have added a Discussion section explicitly addressing how our data predict resting culture performance. (lines 324-332)

Whereas extended antigen-free culture would provide direct demonstration of memory characteristics, multiple lines of evidence support that BATF3-OE cells would show superior performance in such assays: Progressive enrichment over 30-40 days indicates

sustained survival capacity; Memory-associated chromatin accessibility patterns are established and stable; Clinical data demonstrate that BATF3-regulated programs correlate with durable in vivo responses requiring months of persistence; Published studies showed BATF3 promotes memory T cell survival during antigen-free periods.(Ataide et al., 2020) The convergence of molecular, functional, and clinical evidence substantially supports our conclusions about BATF3's role in promoting memory-associated characteristics.

Comments from the Reviewer #2 and reply to the comments

In this study, Umemura et al show that overexpression of BATF3 in engineered CAR-T cells enhances their proliferation. They also find that BATF3-OE CAR T cells drive lower IFN-g and IL-2 secretions. This work supports and successfully repeats the previous findings of Ataide et al (2023), they found that BATF3 overexpression in CD8+T cells induces their survival and transition to a memory phenotype. To investigate the role of BATF3 in memory T cell differentiation, the authors integrated multiple big data analyses based on published datasets and their ATAC-seq to provide some new information about BATF3's essential role in memory T cell differentiation processes. Notably, the original text contains several nonstandard expressions and semantic ambiguities that require careful revision.

Comments to the authors:

Main points:

1) I doubt the accuracy of using "early memory T cells" and "more differentiated T cells" to distinguish Flu-specific T cells and CMV-specific T cells in the text. Although the memory T cell pool is heterogeneous, distinct subsets are well defined based on their cell surface marker expressions. At least, I suggest presenting the data on memory pool dynamics between Flu-T and CMV-T populations. For example, using CD45RO⁺CCR7⁺CD62L⁺, CD45RO⁺CCR7⁻CD62L⁻ and CD45RA⁺CCR7⁻CD62L⁻ to present the proportion of central memory T cells (Tcm), effector memory T cells (Tem) and Terminally Differentiated Effector Memory T Cells (Temra) between two virus-specific T cell groups, respectively.

Response: We fully agree with this comment (which overlaps with Reviewer #1, Comment 1). As detailed in our response to Reviewer #1, Comment 1, we have responded as follows.

Changes: To clarify the premise before explaining the experimental results, we added the following text to the beginning of the first paragraph. We also added two new references. (new references: Appay et al., Nature Medicine 2002. Memory CD8+ T cells vary in differentiation phenotype in different persistent virus infections. and Ibegbu et al. Immunology. 2009. Differential expression of CD26 on virus-specific CD8(+) T cells during active, latent and resolved infection.)

The immune response to cleared acute infections without persistent antigen (such as influenza) is characterized by the long-term presence of true quiescent memory cells, whereas the chronic phase of persistent infections (CMV, EBV) is characterized by the presence of effector phenotypes with varying degrees of more highly differentiated cells. (new references: Appay et al., Nature Medicine 2002. Memory CD8+ T cells vary in differentiation phenotype in different persistent virus infections. and Ibegbu et al. Immunology. 2009. Differential expression of CD26 on virus-specific CD8(+) T cells during active, latent and resolved infection.) (lines 111-114)

To show the phenotypic differences between Flu-specific CTLs and CMV-specific CTLs, we added additional phenotype characterization in new Figure 1D, E, and new Supplemental Figure S1B (representative flow plots).

When we examined the surface phenotype of tetramer-positive cells during the culture period, Flu-specific CTLs were predominantly CD45RA+/CD62L+ naive cells before stimulation, and CD45RA-CD62L+ central memory T cells on day 10 (Figure 1D,E and Supplemental Figure S1B). (lines 121-123)

2) In Figure 2, I suggest showing the validated expression of selected genes between Flu-specific CD8 T cells and CMV-specific CD8 T cells after twice stimulation. Or at least, to provide a rigorous rationale for selecting these 5 candidates out of the list. Because they are not particularly prominent in Fig. 2B.

Response: We apologize for not providing sufficient justification for our candidate gene selection.

Changes: To clarify the selection criteria, we modified as follows.

To clarify, we have added **the \log_2 (fold change) criteria**. (line 137) In fact, we have already mentioned that we have confirmed expression in T cells based on literature and database searches. (lines 137-139)

3) In Fig. 4H and I, if the authors claim that TF+ virus specific T cells have anti-apoptotic traits, the typical apoptosis experiments should be added in the main figures. The differences in TF+/TF- ratio between BATF or BATF3+ and Control CD8 T cells could also be caused by increased proliferation.

Response: We agree with the Reviewer. Since both possibilities are plausible, we added the following sentence to reflect both possibilities.

Changes: Taken together, BATF and BATF3, but not JUN transduction, added certain anti-apoptotic traits to virus-specific T cells **or increased proliferation capacity**. (line 203)

4) Representative flow cytometry plots should be involved in Fig. 5G and H, and the gating strategy should be shown in legends or in figures as well. Otherwise, we don't even know if the authors gate out dead cells or not.

Response: According to the Reviewer's suggestion, we included representative flow plots as a new Figure 5G and revised/renumber other Figure legends. Also we added gate strategy description in the Figure legends. The description "of total cells" in the legend was incorrect, so we corrected it as follows.

Changes: Figure 5G legend: **G, Representative flow plots of TF transduced CD19 CAR-T cells after three repetitive stimulations. Live and lymphocyte were gated, and then CD8+/GFP+ cells were gated to show as CD45RA/CD62L flow plots.** (lines 718-721)

Percentages of TF-positive cells are shown. (line 723)

5) While McCutcheon et al. (2023) have previously reported that BATF3 overexpression enhances the anti-tumor activity of CAR-T cells, it is regrettable that the current manuscript lacks data on the anti-tumor efficacy or cytotoxic activity of BATF-OE, BATF3-OE, or BATF/BATF3-co-OE CAR-T cells. Notably, there is a somewhat conflict in the manuscript: BATF3 was initially identified from a subset of Flu-specific T cells with high IL-2 production, yet the authors subsequently show that BATF3-OE CAR-T cells secrete lower levels of IFN- γ and IL-2. A detailed discussion addressing this discrepancy is warranted.

Response: As the Reviewer pointed out, we initially identified related genes within a cell population with higher IL-2 expression, but ultimately, the correct description is that IL-2 expression is sustained for a longer period. Therefore, we revised the relevant parts as follows.

Changes:

we screened genes associated with a **sustained** IL-2 production capacity in human virus-specific CD8+ T cells and chose several TFs as candidates. (line 293)

We initially focused on BATF3 and other related genes within a cell population with higher IL-2 expression, however, ultimately, the correct description was that IL-2 expression is sustained for a longer period. (lines 306-308)

Minor issues:

1) Line 122: Please clarify if "TFs" is short for "Transcriptional Factors"; Here, "TFs" appears for the first time in the manuscript.

Response: We added expansion for TF as transcription factors.

Change: TFs → **Transcription Factors** (line 130)

2) Line 125: I am confused about "All three donors provided the CMV-CTL and Flu-CTL lines." What do lines refer to here?

Response: We apologize for the confusion. By "lines" we meant "CTL cultures" or "CTL populations." We have revised this sentence to clarify:

Original text (Line 125): All three donors provided the CMV-CTL and Flu-CTL lines.

Changes: Paired CMV-CTL and Flu-CTL cultures were generated from each of the three donors. (lines 134-135)"

3) Line 134: A reference citation is missing for "the recently reported BHLHE40-OE promotes effector differentiation of T cells."

Response: We added two references about BHLHE40 function as follows. We also modified the sentence to describe the BHLHE40 functions more precisely.

Changes: We have modified the sentence as follows.

and *BHLHE40*, which has been reported to regulate T cell effector function and differentiation. (line 144)

Two new references were added.

In vitro modeling of CD8⁺ T cell exhaustion enables CRISPR screening to reveal a role for BHLHE40. Wu JE et al. Science Immunology. 2023 Aug 25; 8(86): eade3369. doi: 10.1126/sciimmunol.ade3369.

The Transcription Factor Bhlhe40 Programs Mitochondrial Regulation of Resident CD8⁺ T Cell Fitness and Functionality. Li C et al. Immunity. 2019 Sep 17;51(3):491-507.e7.doi: 10.1016/j.immuni.2019.08.013. (line 144-145)"

4) Line 143: Please clarify if "TM domain" means "transmembrane domain".

Response: The reviewer is correct. TM means transmembrane domain. We apologize for the confusion. Because transmembrane and costimulatory domain are both derived from CD28, we changed as below.

Changes: CD28 transmembrane and costimulatory domain, (line 155)

5) It will be much better if the authors can summarize the conclusions of Fig.6 and Fig.7 concisely and accurately at the end of their paragraph, respectively.

Response: Thank you for your kind suggestion. We added conclusions of each Figures at the end of paragraphs as below.

Change:

Conclusion for Figure 6: **Together, these data demonstrate that BATF3 overexpression dynamically regulates chromatin accessibility to establish a memory-associated transcriptional program, characterized by upregulation of pro-survival genes and downregulation of exhaustion-associated genes, specifically during the transition from effector to memory phase (day 7 to day 14).** (lines 255-258)

Conclusion for Figure 7: **Collectively, BATF3 and BATF regulate distinct transcription factor networks, with BATF3-associated genes (including KLF6) enriched in responding CAR-T patients, linking BATF3-mediated memory programming to clinical therapeutic outcomes.** (lines 288-291)

6) "Memory signature score" shown in abstract is inappropriate that may cause difficulties in understanding before reading the paper.

Response: We agree that 'memory signature score' is technical terminology that may be unclear without reading the full manuscript. We have revised the abstract to make this more accessible.

Change:

Original text: 'Memory signature score analysis demonstrated significant increases only in BATF3-overexpressing cells...'

Revised text: **Chromatin accessibility analysis revealed that BATF3-overexpressing cells specifically upregulated genes associated with memory T cell characteristics and downregulated exhaustion-associated genes,** (lines 58-60)

Comments from the Reviewer #3 and reply to the comments

This manuscript investigates the role of the AP-1 family transcription factor BATF3 in human virus-specific CD8⁺ T cells and CD19 CAR-T cells. The authors identify BATF3 from RNA-seq of Flu- versus CMV-specific CTLs, test several TF candidates

in virus-specific CTLs and CAR-T cells, perform ATAC-seq with an effector/memory scoring framework, and re-analyze published clinical datasets. The work is extensive and potentially relevant for improving CAR-T persistence, but several interpretations should be toned down and key methods clarified.

Major Comments

1. The central concept that BATF3 is a key intrinsic regulator of CD8⁺ T-cell memory and a promising target to improve adoptive T-cell therapy has already been established by Ataide et al. (Nat Immunol 2020) and further supported in CAR-T cells by McCutcheon et al. (2023). These studies showed that BATF3 promotes the generation and maintenance of memory CD8⁺ T cells and enhances the durability of T-cell-based therapies. Thus, the present manuscript is not conceptually groundbreaking, but provides an incremental yet meaningful extension of this framework.

The added value of the current work lies mainly in:

- (i) the use of human Flu- and CMV-specific CTLs as a physiologically relevant model to screen TFs associated with less- versus more-differentiated CD8⁺ T-cell states;**
- (ii) a systematic comparison of BATF3 and BATF overexpression in both virus-specific CTLs and CD19 CAR-T cells;**
- (iii) integration of ATAC-seq with effector/memory scoring to describe a BATF3-associated, memory-like chromatin accessibility program;**
- (iv) exploratory re-analysis of clinical CAR-T transcriptomic datasets linking BATF3-related gene modules to treatment response.**

I therefore view the novelty as incremental but acceptable, provided that the authors more clearly delineate in the Introduction and Discussion what is genuinely new relative to previous BATF3 studies and soften statements implying that BATF3 is identified here de novo as a master regulator of memory or CAR-T persistence.

Response: We appreciate the Reviewer's acknowledgment that our work provides "incremental yet meaningful extension" of the BATF3 framework. We agree that we should more clearly delineate our novel contributions. We have revised the Introduction to explicitly state: Since BATF3 was not newly discovered by us, we changed "identified" to "focused on."

Change: identified → focused on (line 100)

2. In Figure 1, the study treats Flu-specific CTLs as "earlier memory" and CMV-specific CTLs as more terminally differentiated. The supporting evidence is mainly functional (expansion kinetics and IL-2 production), without full phenotypic characterization. If feasible, adding a simple summary figure or table of the surface phenotype (e.g. CD45RA/RO, CCR7) of Flu- versus CMV-CTLs would strengthen this point.

Response: We fully agree with this comment (which overlaps with Reviewer #1, Comment 1 and Reviewer #2 Comment 1). As detailed in our response to Reviewer #1, Comment 1, we have responded as follows.

Changes:

To show the phenotypic differences between Flu-specific CTLs and CMV-specific CTLs, we added additional phenotype characterization in new Figure 1D, E, and new Supplemental Figure S1B (representative flow plots).

When we examined the surface phenotype of tetramer-positive cells during the culture period, Flu-specific CTLs were predominantly CD45RA⁺/CD62L⁺ naive cells before stimulation, and CD45RA⁻CD62L⁺ central memory T cells on day 10 (Figure 1D,E and Supplemental Figure S1B). (lines 121-123)

3. In Figure 2, the authors ultimately focus on five candidate genes (HTATSF1, IKZF4, BATF3, PHTF2 and ID2), selected from those that are more highly expressed in Flu-CTLs and reported to be expressed in T cells. It is not entirely clear why these five were prioritized over others. It would be helpful if the authors could briefly explain the additional criteria used for this selection (e.g. consistency across donors,

magnitude of fold change, prior functional annotation), and indicate which of these genes are established transcription factors or chromatin regulators versus less well-characterized factors.

Response: We apologize for not providing sufficient justification for our candidate gene selection.

Changes: To clarify the selection criteria, we modified as follows. Also we added one sentence to indicate which of these genes are established TF or chromatin regulators versus less well-characterized factors.

To clarify, we have added **the log₂ (fold change) criteria**. (line 137) In fact, we have already mentioned that we have confirmed expression in T cells based on literature and database searches. (lines 137-139)

Among these five genes, *BATF3*, *IKZF4*, and *ID2* are established TF/regulators with relatively well-characterized roles in T cell biology, whereas *PHTF2* and *HTATSF1* represent less well-characterized factors. (lines 148-150)

4. In Figure 3, the authors note that viral transduction efficiencies differ among the TF constructs (e.g. varying EGFP⁺ frequencies in Fig. 3C). This raises the question of how much of the observed differences in proliferation, persistence and cytokine production might be influenced by unequal TF expression levels rather than by qualitative effects of the TFs themselves.

Response: We have revised the text to clarify that the key finding is the progressive enrichment of BATF3⁺ cells after three rounds of stimulation, rather than the initial transduction efficiency differences among constructs.

Changes: **whereas only BATF3⁺CD19CAR-T cells demonstrated the progressive enrichment of transduced cells** (lines 163-164)

5. In Fig. 5G-H, CD19 CAR-T cells overexpressing BATF or BATF3 display significantly lower T_{CM} and higher T_{EM} proportions after repetitive stimulation. Phenotypically, this shift appears more consistent with an

effector/effector-memory-like state rather than an increase in classical central memory cells. It would therefore be helpful if the authors could address this point in the Results and Discussion, for example by clarifying whether the T_EM population observed here still retains key memory/persistence features and how this phenotype fits with the claim that BATF3 (and BATF) promote "memory" formation.

Response: These data demonstrate that BATF and BATF3 overexpression does not inhibit effector cell differentiation; rather, both TFs initially promote effector differentiation. We have added a sentence to clarify this point.

Changes: **BATF and BATF3-OE does not inhibit effector cell differentiation; rather, both TFs initially promote effector differentiation.** (lines 340-342)

6. In Fig. 7A (and Supplemental Fig. S5C), there appears to be some tension between the "memory correlation score" and the established biology of several transcription factors. Classical memory/differentiation-associated TFs such as TCF7, PRDM1 and ZEB2 are positioned closer to the "effector-side" of the memory score, whereas MAF, MXD4 and KLF6 fall toward the "memory-side". This raises the question of whether the memory score derived from your signatures fully aligns with conventional definitions of memory programs. It would be helpful if the authors could explicitly discuss this point, clarifying that the score reflects a specific gene-signature-based metric and may capture particular stages or types of memory differentiation, rather than a universal "memory vs effector" axis.

Response: This is an insightful observation highlighting the complexity of memory programming. We have added detailed discussion as follows.

Changes:

The positioning of classical differentiation-associated transcription factors in our memory correlation score analysis (Fig 7A) requires clarification. Our memory score is a computational metric derived from single-cell transcriptome data comparing effector versus memory CD8+ T cell states during a specific transition phase, rather than a universal definition of memory programs. Classical TFs like TCF7, PRDM1, and ZEB2

have dynamic, stage-specific roles: TCF7 is critical for establishing memory precursors during initial effector responses but may be transiently elevated at day 7 before decreasing by day 14 (Fig 7B-C), consistent with a transition role rather than memory maintenance; PRDM1 has dual roles in promoting effector function while preventing exhaustion, with optimal memory requiring its downregulation after initial effector phase; and ZEB2 drives effector differentiation but can promote terminal differentiation when chronically elevated. Our signature specifically captures the day 7 to 14 transition under repetitive CAR-T stimulation, which represents memory maintenance under chronic antigen exposure rather than initial memory formation after acute infection. Conversely, factors positioned toward the memory side—MAF (opposes terminal differentiation), MXD4 (antagonizes Myc-driven effector programs), and KLF6 (our novel finding enriched in responding CAR-T patients, Fig 7D-E)—may be particularly important for memory maintenance under chronic stimulation conditions. Despite counterintuitive positioning of some classical factors, the score's biological relevance is validated by series of our experimental data: specific increases only in BATF3-OE cells (Fig 6D); functional memory characteristics of BATF3-OE cells (survival, low exhaustion markers); clinical correlation with CAR-T response (Fig 7D-E). Thus, we believe that our score reflects a context-specific metric capturing memory programming relevant to CAR-T persistence under chronic stimulation, rather than a universal classification system.

Minor Comments

1. In the current version, Supplemental Fig. S6 appears to show a volcano plot of responders vs non-responders, whereas the legend describes "temporal expression patterns of individual genes", which does not match the figure. Please correct the legend and clearly state which dataset (Tisagen vs axi-cel) is shown. I also recommend carefully checking all other figures, legends and dataset labels to ensure there are no similar inconsistencies.

Response and changes: We apologize this mistake. We corrected to accurately describe the volcano plot content. Changed from "temporal expression patterns" to "Reanalysis of

single-cell RNA-seq data from patients treated with Axicabtagene ciloleucel. Volcano plot showing differentially expressing genes between responders (R) and non-responders (NR).

We reviewed all Figures and legends and confirmed consistency. Figure 7D shows Tisagenlecleucel data, Supplementary Figure S6 show Axicabtagene ciloleucel data.

2. Throughout the manuscript, the authors frequently use an adjusted P value threshold of 0.1 to define differentially expressed genes and to select candidates. This cutoff is relatively lenient compared with the more conventional threshold of 0.05. I would recommend either tightening the statistical threshold (e.g. adj. $P < 0.05$) where feasible, or explicitly justifying the choice of 0.1 (for example, as a screening criterion given the sample size and exploratory nature of the analysis).

Response: Added justification in Methods (lines 450-456)

Changes:

For differential expression analysis, we used an adjusted P value threshold of 0.1, which is appropriate for an exploratory screening phase given our sample size (n=3 donors per group). This threshold balances the risk of false negatives (missing true candidate genes) against false positives, with the understanding that all candidates would be subsequently validated functionally. The use of $P < 0.1$ for discovery-phase transcriptomic screening with subsequent functional validation is an accepted approach in the field (Ge et al., BMC Bioinformatics 2018). (lines 450-456)

3. At lines 134-135, the manuscript states that "BHLHE40, on which the recently reported BHLHE40-OE promotes effector differentiation of T cells", but no supporting reference is provided.

Response: We added two references about BHLHE40 function as follows. We also modified the sentence to describe the BHLHE40 functions more precisely.

Changes: We have modified the sentence as follows.

and BHLHE40, which has been reported to regulate T cell effector function and differentiation. (line 144)

Two new references were added.

In vitro modeling of CD8⁺ T cell exhaustion enables CRISPR screening to reveal a role for BHLHE40. Wu JE et al. *Science Immunology*. 2023 Aug 25; 8(86): eade3369. doi: 10.1126/sciimmunol.ade3369.

The Transcription Factor Bhlhe40 Programs Mitochondrial Regulation of Resident CD8⁺ T Cell Fitness and Functionality. Li C et al. *Immunity*. 2019 Sep 17;51(3):491-507.e7.doi: 10.1016/j.immuni.2019.08.013. (lines 144-145)"

4. At lines 169-171, the authors state that "We selected 22 AP-1 family TFs and reanalyzed the comparison data between day 20 CMV-CTL and day 20 Flu-CTL and found that FOSL1 and BATF were also weakly significant (Fig. 4A)." However, it is not explained how these 22 AP-1 family members were defined or selected. Please clarify the criteria and source used to compile this AP-1 TF list (e.g. specific database, gene ontology term, or prior publications) .

Response: We added a reference: Chinenov & Kerppola, *Oncogene* 2001, which provides the canonical list of AP-1 family transcription factors used for our analysis.

Change: We put a reference (Added reference: Chinenov & Kerppola, *Oncogene* 2001, which provides the canonical list of AP-1 family transcription factors used for our analysis). (line 183)

We thank the Editor and Reviewers for their careful reviews and insightful comments and trust that this manuscript is now suitable for publication in *Life Science Alliance*.

Yours sincerely,

Seitaro Terakura, M.D., Ph.D.
Department of Hematology and Oncology,
Nagoya University Graduate School of Medicine

Nagoya, Japan

Tel: +81-52-744-2145

Fax: +81-52-744-2161

E-mail: tseit@med.nagoya-u.ac.jp

January 22, 2026

RE: Life Science Alliance Manuscript #LSA-2025-03550-TR

Dr. Seitaro Terakura
Nagoya University
Department of Hematology and Oncology
65 Tsurumai-cho
Showa-ku
Nagoya, Aichi 466-8550
Japan

Dear Dr. Terakura,

Thank you for submitting your revised manuscript entitled "BATF3 regulates differentiation of CD8+ T lymphocytes and memory differentiation program". We returned this manuscript to Reviewers 1 and 2, whose comments are below. As you will see Reviewer 2 is concerned by the differential gene expression analysis in Figure 2, an issue that was also noted by Reviewer 3. It is not surprising that T cells from different donors, analyzed in the paired method as done here, may result in overall few genes whose expression was consistently altered across samples. While we appreciate the concern and the rigor of Reviewer 2, we feel that readers can evaluate the strength of this conclusion from the data provided and subsequent analyses. We leave the addition of GO/KEGG analyses as suggested by this reviewer to your discretion. However this potential limitation and the rationale for this analysis, which is stated in the methods section, should be stated in the results section. We would be happy to publish your paper in Life Science Alliance pending this change and final revisions necessary to meet our formatting guidelines.

MANUSCRIPT ORGANIZATION AND FORMATTING:

To avoid unnecessary delays in the acceptance and publication of your paper, please read the following information carefully. Full guidelines are available on our Instructions for Authors page, <https://www.life-science-alliance.org/authors>

- Please upload all figure files individually, including the supplementary figure files; all figure legends should only appear in the main manuscript file.
- Please add ORCID ID for corresponding (and secondary corresponding) author--you should have received instructions on how to do so.
- Please add a Category for your manuscript in our system.
- Please add the X and Bluesky handles of your host institute/organization, as well as your own and/or one of the authors, in our system.
- Please remove information about the word counts and the following information from the title page in the manuscript file.
- The abstract should be a single paragraph not exceeding 175 words and must match between the system and the manuscript file.
- In the Methods section, please indicate the concentrations for all antibodies used for flow cytometry. Please also remove references to Metascape and WebGestalt analyses of bulk RNA seq, as it appears these were not applied to these data.
- Please update the Data Availability statement with an accession number for the ATAC-seq dataset. Please also deposit the bulk RNA seq data in GEO add an accession number for this dataset. These are required prior to publication.
- Please rename the section "Declaration of Interests" to "Conflict of Interests."
- Please remove the Financial Support section. This information is to be provided in our submission system only, and will be visible to readers of this work on our website.
- Please move your main and supplementary figure legends in the main manuscript text after the references section.
- Please remove legends from the supplementary figures.
- Please use the [10 author names et al.] format in your references (i.e., limit the author names to the first 10).
- Please add callouts for Figures 6J and S4A-D to your main manuscript text.
- The 4 flow cytometry plots in the Day 30 row in Figure 4E appear to be duplicated in Supplemental Figure 3, but with different percentages listed. Please verify that the flow plots in both figures are correct. If these two figures should show the same flow plots, please correct the percentages and note the duplication in the legend for each figure.

LSA encourages authors to provide a 30-60 second video where the study is briefly explained. We will use these videos on social media to promote the published paper and the presenting author (for examples, see <https://docs.google.com/document/d/1-UWCfbE4pGcDdcgzcmiuJl2XMBJnxKYeqRvLLrLSo8s/edit?usp=sharing>). Corresponding or first-authors are welcome to submit the video. Please submit only one video per manuscript. The video can be emailed to contact@life-science-alliance.org

FINAL FILES:

The following items are required for acceptance.

The license to publish form must be signed before your manuscript can be sent to production. A link to the license to publish form will be available to the corresponding author only. Please take a moment to check your funder requirements.

Thank you for your attention to these final processing requirements. Please revise and format the manuscript and upload materials as soon as you are able.

Thank you for this interesting contribution to the literature. We look forward to publishing your paper in Life Science Alliance.

Sincerely,

Reviewer #1 (Comments to the Authors (Required)):

My concerns have been addressed.

Reviewer #2 (Comments to the Authors (Required)):

I appreciate the authors' efforts in making the revisions. Although the novelty of this study has already been reported early before, it still provides some new and valuable information on the essential role of BATF3 in memory T cell differentiation processes in human cells to make this study to be a good candidate for publication in Life Science Alliance. I still have some suggestions for presenting the data of Figure 2B (please refer to my major point 2, I believe other Reviewer also has this concern). I understand that a threshold of FDR < 0.1 and fold-change > 2 is acceptable in DESeq2 data package

analysis, but honestly, this selection condition is extremely loose. Ge et al. (Ge et al., BMC Bioinformatics, 2018) identified ~1000 up- and down-regulated genes by using this threshold. Hence, using the selected gene list to perform a GO or KEGG network analysis is necessary and helpful in narrowing down the target gene candidates/pathway. Perhaps the authors could try a heatmap to show the frequencies of remarkable genes (concluding HTATSF1, IKZF4, BATF3, PHTF2, and ID2) in such an enriched pathway according to GO/KEGG analyses. Unless there are too few genes to perform pathway screening. In fact, my issue is that picking HTATSF1, IKZF4, BATF3, PHTF2, and ID2 out of the list seems quite forced; they differ significantly from the top-ranked genes based on absolute fold change.

Besides, I appreciate the newly added Figure 1D, E, and Figure S1B for addressing my major point 1. Due to Figure S1B shows somewhat blurry representative plots of gating Tcm, adding CCR7 and CD45RO in the staining panel could be helpful for a clear-cut edge during gating different populations. However, compared to the plots on day 0, these results presented the increased trend of Tcm in Flu-CTL on day 10 and supported the observation of "Flu-CTL seemed to be less differentiated T cell". Therefore, I think it's not precise but acceptable not to make any additional modifications.

Overall, except for my major concern 2, the rest of my issues have been addressed.

March 30, 2026
Tim Fessenden, PhD
Scientific Editor
Life Science Alliance

Dear Dr. Fessenden,

We sincerely appreciate the careful review by the Editors and the Reviewers of our manuscript (LSA-2025-03550-TR), and we have revised our manuscript accordingly.

The following points below address the comments by the Reviewers. To facilitate your consideration, **the pertinent reviewers' comments are highlighted in bold**, and the response follows in regular font. Changes in the manuscript are highlighted in red.

Comments from the Reviewer #2 and reply to the comments

I appreciate the authors' efforts in making the revisions. Although the novelty of this study has already been reported early before, it still provides some new and valuable information on the essential role of BATF3 in memory T cell differentiation processes in human cells to make this study to be a good candidate for publication in Life Science Alliance.

I still have some suggestions for presenting the data of Figure 2B (please refer to my major point 2, I believe other Reviewer also has this concern). I understand that a threshold of FDR < 0.1 and fold-change > 2 is acceptable in DESeq2 data package analysis, but honestly, this selection condition is extremely loose. Ge et al. (Ge et al., BMC Bioinformatics, 2018) identified ~1000 up- and down-regulated genes by using this threshold. Hence, using the selected gene list to perform a GO or KEGG network analysis is necessary and helpful in narrowing down the target gene candidates/pathway. Perhaps the authors could try a heatmap to show the frequencies of remarkable genes (concluding HTATSF1, IKZF4, BATF3, PHTF2, and ID2) in such an enriched pathway according to GO/KEGG analyses. Unless there are too few genes to perform pathway screening. In fact, my issue is that picking HTATSF1, IKZF4, BATF3, PHTF2, and ID2 out of the list seems quite forced; they differ

significantly from the top-ranked genes based on absolute fold change.

Besides, I appreciate the newly added Figure 1D, E, and Figure S1B for addressing my major point 1. Due to Figure S1B shows somewhat blurry representative plots of gating Tcm, adding CCR7 and CD45RO in the staining panel could be helpful for a clear-cut edge during gating different populations. However, compared to the plots on day 0, these results presented the increased trend of Tcm in Flu-CTL on day 10 and supported the observation of "Flu-CTL seemed to be less differentiated T cell".

Therefore, I think it's not precise but acceptable not to make any additional modifications.

Overall, except for my major concern 2, the rest of my issues have been addressed.

We sincerely thank Reviewer #2 for thoughtful comments and constructive suggestions.

We are pleased that the reviewer finds our study provides new and valuable information on BATF3's essential role in human memory T cell differentiation.

Response: We appreciate the reviewer's rigorous evaluation and understand the concern regarding our gene selection approach. We acknowledge that our threshold (FDR < 0.1 and fold-change > 2) is relatively permissive for an exploratory screening phase.

As you noted, the paired analysis method used here with T cells from different donors may inherently result in fewer genes with consistently altered expression across samples due to donor-to-donor variability. This biological reality reflects the heterogeneity of human immune responses, which differs from more homogeneous experimental systems. Regarding GO/KEGG pathway analysis, we considered performing GO/KEGG enrichment analysis as suggested. However, given the relatively small number of consistently altered genes identified (258 DEGs on day 20, 204 DEGs on day 30), and considering that many genes lost significance between day 20 and day 30 (as shown in Figure 2C), we are concerned that formal pathway enrichment analysis may not yield statistically robust results with this dataset.

Regarding our gene selection strategy, we acknowledge that our selection of HTATSF1, IKZF4, BATF3, PHTF2, and ID2 was not solely based on absolute fold-change ranking. Our selection criteria prioritized:

1. Genes with established or potential roles as transcription factors/regulators in T cell biology.
2. Consistency of differential expression between day 20 and day 30.
3. Biological plausibility for involvement in memory T cell formation.

While BATF3, IKZF4, and ID2 are relatively well-characterized transcription factors/regulators in T cell biology, PHTF2 and HTATSF1 represent less-characterized factors that warranted investigation. Importantly, subsequent functional validation experiments (Figures 3-6) confirmed that BATF3 OE significantly impacts T cell proliferation, cytokine production, and memory differentiation, supporting the validity of our candidate gene approach.

Following the Scientific Editor's suggestion, we have added text to the Results section that explicitly states the potential limitations of this analysis and the rationale for our approach (which was previously only described in the Methods section). This addition provides readers with the necessary context to evaluate the strength of our conclusions.

Accordingly we revised text and added to Results section as follows (lines 115-118)

Changes: **Given the inherent donor-to-donor variability in human T cell responses, our paired analysis approach with a permissive threshold (FDR < 0.1) was designed to prioritize sensitivity in identifying candidate genes for subsequent functional validation, accepting a higher risk of false positives during this discovery phase.**

We thank the Editor and Reviewers for their careful reviews and insightful comments and trust that this manuscript is now suitable for publication in *Life Science Alliance*.

Yours sincerely,

Seitaro Terakura, M.D., Ph.D.
Department of Hematology and Oncology,

Nagoya University Graduate School of Medicine
Nagoya, Japan
Tel: +81-52-744-2145
Fax: +81-52-744-2161
E-mail: tseit@med.nagoya-u.ac.jp

March 31, 2026

RE: Life Science Alliance Manuscript #LSA-2025-03550-TRR

Dr. Seitaro Terakura
Nagoya University
Department of Hematology and Oncology 65 Tsurumai-cho Showa-ku Nagoya
65 Tsurumai-cho
Showa-ku
Nagoya, Aichi 466-8550
Japan

Dear Dr. Terakura,

Thank you for submitting your Research Article entitled "BATF3 regulates differentiation of CD8+ T lymphocytes and memory differentiation program". It is a pleasure to let you know that your manuscript is now accepted for publication in Life Science Alliance. Congratulations on this interesting work.

DISTRIBUTION OF MATERIALS:

Again, congratulations on a very nice paper. I hope you found the review process to be constructive and are pleased with how the manuscript was handled editorially. We look forward to future exciting submissions from your lab.

Sincerely,
